# Incorporating moisture content in surface energy balance modeling of a debris-covered glacier

Alexandra Giese[1], Aaron Boone[2], Patrick Wagnon[3], and Robert Hawley[1]

[1]Department of Earth Sciences, Dartmouth College, Hanover NH USA
[2]CNRM-GAME - Groupe d'Étude de l'Atmosphère Météorologique, Toulouse FRANCE
[3]Univ. Grenoble Alpes, CNRS, IRD, Grenoble-INP, IGE, 38000 Grenoble FRANCE

**Correspondence:** Alexandra Giese (algiese@gmail.com)

**Abstract.** Few surface energy balance models for debris-covered glaciers account for the presence of moisture in the debris, which invariably affects the debris layer's thermal properties and, in turn, the surface energy balance and sub-debris melt of a debris-covered glacier. We adapted the Interactions between Soil, Biosphere, and Atmosphere (ISBA) land surface model within the SURFace EXternalisée (SURFEX) platform to represent glacier debris rather than soil. The new ISBA-DEBris model includes the varying content, transport, and state of moisture in debris with depth and through time. It robustly simulates not only the thermal evolution of the glacier-debris-snow column but also moisture transport and phase changes within the debris – and how these, in turn, affect conductive and latent heat fluxes. We discuss the key developments in the adapted ISBA-DEB and demonstrate the capabilities of the model, including how the time- and depth-varying thermal conductivity and specific heat capacity depend on evolving temperature and moisture. Sensitivity tests emphasize the importance of accurately constraining the roughness lengths and surface slope. Emissivity, in comparison to other tested parameters, has less of an effect on melt. ISBA-DEB builds on existing work to represent the energy balance of a supraglacial debris layer through time in its novel application of a land surface model to debris-covered glaciers. Comparison of measured and simulated debris temperatures suggests that ISBA-DEB includes some – but not all – processes relevant to melt under highly permeable debris. Future work, informed by further observations, should explore the importance of advection and vapor transfer in the energy balance.

## 1   Introduction

Enhancing the melt of underlying ice when thin and inhibiting it when thick (Östrem, 1959), supraglacial debris is known to affect the surface energy balance and retreat patterns of mountain glaciers. Supraglacial debris covers 11% of glacier area in High Mountain Asia (HMA) (Kraaijenbrink et al., 2017), a region that contains the highest volume of ice on Earth outside the polar regions and where glacier melt flows into rivers that deliver water to 800 million people (Pritchard, 2019). Understanding sub-debris melt is crucial for making informed projections of climate change impacts and associated water security issues in HMA.

Sub-debris ablation is fundamentally a function of the temperature at the surface of the debris and the ability of the debris to conduct heat to its base at the ice-debris interface. Therefore, the amount of ice melt under debris is determined by local

| Debris porosity ($\phi$) = 0.39 | Thermal Conductivity (W m$^{-1}$K$^{-1}$) | Specific Heat Capacity (J kg$^{-1}$K$^{-1}$) | Density (kg m$^{-3}$) | Volumetric Heat Capacity (J m$^{-3}$K$^{-1}$) | Diffusivity (m$^2$ s$^{-1}$) |
|---|---|---|---|---|---|
| **Dry debris** | **0.94** | **948** | **1690** | **1602120** | **5.867 $\times 10^{-7}$** |
| Air | 0.024 | 1000 | ∼0.67 | - | - |
| **Water-saturated debris** | **1.16** | **-** | **-** | **3247140** | **3.572 $\times 10^{-7}$** |
| Water | 0.57 | 4218 | 1000 | - | - |
| **Ice-saturated debris** | **1.81** | **-** | **-** | **2356700** | **7.680 $\times 10^{-7}$** |
| Ice | 2.2 | 2110 | 917 | - | - |

**Table 1.** Thermally relevant properties of dry debris, in which interstitial pore spaces are filled with air; water-saturated debris; and ice-saturated debris of porosity ($\phi$) = 0.39. References and calculation details are given in a complementary table in the Supplement (Table S1). Air density is a function of elevation, air temperature, and air moisture. Thermal conductivity presented by Reid and Brock (2010) is an "effective" value, from measurements, that is a function of debris' unspecified porosity and any moisture content at the time of measurement (Collier et al., 2016). Brock et al. (2010) used a published value of specific heat (948 J kg$^{-1}$K$^{-1}$). We assume that these values of thermal conductivity and heat capacity (here listed for "dry debris") are valid for dry debris on West Changri Nup glacier and subsequently perform sensitivity tests. More details about this assumption and its implications can be found in the Discussion.

meteorological conditions and physical properties of the debris itself. The efficiency with which a debris layer conducts heat is determined by its thermal diffusivity $\kappa$ (m$^2$ s$^{-1}$). $\kappa$ is given by thermal conductivity K (W m$^{-1}$K$^{-1}$) normalized by the volumetric heat capacity (J m$^{-3}$K$^{-1}$), itself a product of density $\rho$ (kg m$^{-3}$) and heat capacity $c$ (J kg$^{-1}$K$^{-1}$). Debris properties beyond thickness are inherently difficult to constrain; a debris layer is comprised of rock clasts of different sizes, angularities, and lithologies that are distributed and sorted heterogeneously over the ablation zone. A debris layer's interstitial spaces may be comprised of air or percolating water, which itself undergoes phase changes as a function of temperature.

Moisture has been largely unaddressed in glacier models, despite the fact that water and ice affect the thermal properties of a debris layer. Table 1 contrasts bulk thermal conductivity, heat capacity, and density of dry debris with debris of the same porosity ($\phi$ = 0.39) that has water-filled and ice-filled interstitial spaces. A number of studies (e.g. Conway and Rasmussen, 2000; Reznichenko et al., 2010; Nicholson and Benn, 2012; Collier et al., 2014) have emphasized the importance of moisture to the thermal properties of debris, particularly in transition seasons. Rounce and McKinney (2014) found a dramatic increase in conductivity from the top 10 cm of debris on Khumbu region glaciers to the deeper depths; they attribute this difference to water content, noting that Nicholson and Benn (2006) found the conductivity of fully saturated debris to be a factor of 2 − 3 greater than that of dry. Importantly, water content shows an association with grain size, too: coarser sediments are less likely to have wet surfaces because fine-grained sediment has small void spaces and, thus, greater capacity for water retention (Juen et al., 2013; Blum et al., 2018).

Further, evaporation and sublimation will lower the surface temperature and remove mass from the system. Condensation and deposition have the opposite effect. Sakai et al. (2004) suggest that neglecting evaporation in energy balance computations can cause an overestimation of sub-debris melt rates by a factor of two.

Most existing models have assumed a dry debris layer, with rain, snowmelt, and glacier melt running off instantaneously (e.g. Lejeune et al., 2013; Rounce and McKinney, 2014). The few studies that do address moisture focus on end member cases (Nakawo and Young, 1981; Nicholson and Benn, 2006), explicitly account for moisture only when relative humidity is 100% (Reid et al., 2012; Fyffe et al., 2014), incorporate a thickness-dependent "wetness factor" (Fujita and Sakai, 2014), or parameterize latent heat based on relative humidity and rain (Rounce et al., 2015). Evatt et al. (2015) advanced debris-covered glacier modeling by accounting for the evaporative heat flux at the base of the debris and, in doing so, the wind speed above and within a debris layer. Including the thickness-dependent wind dynamics in their energy balance model contributed to their reproduction of Östrem (1959)'s thickness–ablation curve. However, their model does not account for moisture beyond that which is evaporated; like other the models, it assumes that melt runs off and does not affect the system's energy (except in the case of evaporation).

Collier et al. (2014) introduced the first energy-balance model that included an evolving, partially saturated debris layer. The model treated moisture through a reservoir approach and calculated the water vapor partial pressure gradient to inform calculations of latent heat fluxes within the debris. This study laid the groundwork for modeling moisture and identified the need for a physically-based approach to incorporating vertical transport processes (i.e. capillary action, hydraulic gradient-driven flow, etc.) and to prognosing the distribution and phase changes of moisture with depth and through time.

Here, we introduce a model that, to our knowledge, is the first to incorporate moisture with consideration of its vertical transport processes and distribution in debris; ISBA-DEB is capable of representing vertical moisture fluxes, phase changes, and moisture retention. We adapt the Interactions between Soil, Atmosphere, and Biosphere (ISBA) soil model housed within the SURFace EXternalisée (SURFEX) platform of Météo-France to include boundary conditions, thermal properties, hydraulic properties, and runoff parameterizations appropriate for supraglacial debris. The ISBA-DEB model is capable of solving not only the heat equation but also moisture transport and retention via the mixed-form Richards' equation.

In this paper, we show capabilities of the model, evaluate its performance, and conduct a series of sensitivity tests on input parameters. We ran ISBA-DEB by driving it with two years of gap-filled *in situ* meteorological data from West Changri Nup glacier and compared output to debris measurements over the same period.

We highlight the important physical processes that need to be accounted for in any debris-covered glacier melt model, such as conduction and phase change of water and ice in the debris. We also discuss the limitations of our model and propose some further considerations for making improvements.

## 2  Field Site and Data

### 2.1  Field Site: West Changri Nup Glacier

West Changri Nup glacier (Figure 1, 27.97 $^o$N, 86.76 $^o$E), also known as White Changri Nup glacier, has an area of 0.92 km$^2$ (measured 2013), ranges in elevation 5330 – 5690 m, and has a small debris-covered area despite being mostly composed of clean ice. The debris is a granitic metamorphic mix, likely consisting of gneissic clasts eroded from the surrounding cliffs (Searle et al., 2003); see Lejeune et al. (2013) for further details and a photograph of the debris. West Changri Nup lies 200 m southwest of North Changri Nup glacier (Sherpa et al., 2017; Vincent et al., 2016) in the Mt. Everest region of Nepal. The ablation zone of North Changri Nup glacier is dominated by a debris cover that has an insulating effect on mass balance (Vincent et al., 2016). Ice cliffs, despite imparting a localized ablation rate of ∼3 times that of the glacier tongue, do not compensate for the ablation reduction impact of the debris on North Changri Nup glacier (Brun et al., 2018). Field measurements and observations confirm the presence of water in debris: density measurements at four sites show that deeper debris retains more moisture, and water has been observed to both wet the debris and pool within it.

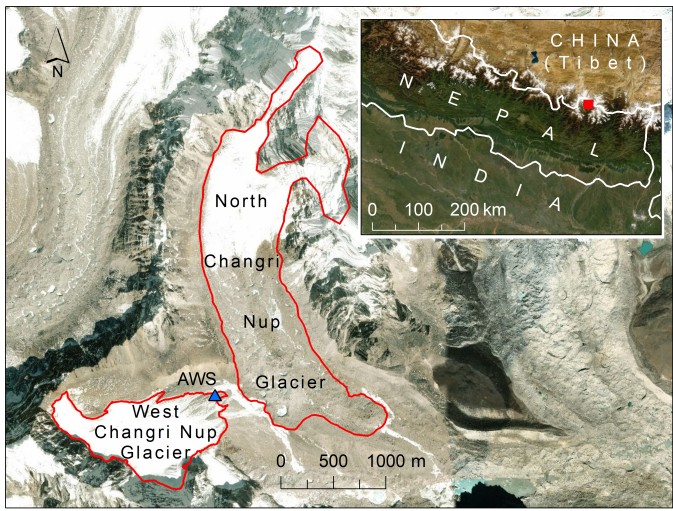

**Figure 1.** A map of West Changri Nup and North Changri Nup glaciers, showing the location of the AWS (Section 2.2), which is also the location of the measurements of debris temperature and point mass balance. Source: Esri, DigitalGlobe, GeoEye, Earthstar Geographics, CNES/Airbus DS, USDA, USGS, AeroGRID, IGN, and the GIS User Community. The glacier outlines are from Sherpa et al. (2017).

### 2.2  In Situ Measurements

An AWS located at 5360 m a.s.l. on a 0.03 km$^2$ debris-covered area of West Changri Nup glacier (Sherpa et al., 2017; Vincent et al., 2016, Figure 1) supplied the meteorological driving data for the model. The AWS was also the site of debris temperature and point mass balance measurements used for model calibration and validation. Half hourly meteorological measurements

6 December 2012 15:00 – 28 November 2014 13:30 local Nepal time provided all necessary data to force the model, and additional half hourly data included ultrasonic depth from an SR50 adjacent to the AWS. During the December 2012 – November 2014 period used for this study, there were four resistance temperature probes installed at distributed depths (5, 7.5, 10, and 12.5 cm) in the 12.5 cm thick debris 40 cm from the AWS. The bottom temperature sensor was placed at the debris-ice

interface.

In addition to the ultrasonic depth gauge installed at the AWS directly above the resistance temperature probes, there was an 8-m long bamboo mass balance stake installed 5 m uphill (west) of the AWS, in ice covered by ∼10 cm debris. Field campaigns supplied additional measurements of debris density and porosity local to the AWS: the mass of debris samples in filled, known-volume buckets provided the density, and water volume filling interstitial pore spaces gave the porosity. Quantities used for

modeling are averages of 9 samples. The measurements for each sample give means (standard deviations) of 1691 (65) $\text{kg m}^{-3}$ and 0.37 (0.04) for density and porosity, respectively.

The nearest direct measurement of precipitation is from a Geonor T200B all-weather sensor at Pyramid Research Station (5035 m a.s.l., 4.3 km southeast of the AWS). Sherpa et al. (2017) contains details on the precipitation data acquisition and correction; the dataset begins 6 December 2012 and extends beyond the end of our study. We assumed corrected total precip-

itation at Pyramid Research Station equivalent to total precipitation on West Changri Nup glacier. Because of differences in elevation and local microclimates, we repartitioned phase based on AWS local temperature following Wagnon et al. (2009), with subsequent, first-order adjustments to the phase to match the timing of major snowfall events detected by the SR50. Table 2 summarizes available data from these stations and indicates which drive the model.

| Quantity | Data Gaps (%) | Instrument | Accuracy according to the manufacturer |
|---|---|---|---|
| **West Changri Nup AWS (5360 m a.s.l.):** | | | |
| Air temperature* § (°C) | 24 | Temperature and relative humidity probe (Vaisala HMP45C) | ±0.2°C |
| Relative humidity* § (%) | 24 | Temperature and relative humidity probe (Vaisala HMP45C) | ±2% |
| Wind direction (°) and speed § (m s$^{-1}$) | 43 | Anemometer (Young 05103-5) | ±3° and ±0.3 m/s |
| Incident shortwave radiation § (W m$^{-2}$) | 18 | Net radiometer (Kipp and Zonen CNR4) | ±3% |
| Reflected shortwave radiation (W m$^{-2}$) | 18 | Net radiometer (Kipp and Zonen CNR4) | ±3% |
| Incoming longwave radiation § (W m$^{-2}$) | 24 | Net radiometer (Kipp and Zonen CNR4) | ±3% |
| Outgoing longwave radiation (W m$^{-2}$) | 24 | Net radiometer (Kipp and Zonen CNR4) | ±3% |
| Debris temperature @ 5, 7.5, 10, and 12.5 cm # (°C) | 24 | Resistance temperature probes (TCA PT100) | ±0.1°C |
| Ablation/accumulation # (m) | 33 | Ultrasonic depth gauge (Campbell SR50A), Mass balance stake | ±1 cm |
| **Pyramid Research Station (5035 m a.s.l.):** | | | |
| Precipitation (mm w.e.) § | 0 | Geonor T200B | ±0.1 mm |

**Table 2.** Meteorological quantities and debris characteristics measured by the AWS on West Changri Nup glacier 6 December 2012 9:15 – 28 November 2014 7:45 UTC. All sensors give values every 30 minutes; these values are 30 minute averages of data with 30 second scanning intervals for all values except the ultrasonic SR50 and wind direction, which are sampled every 30 min. The Kipp and Zonen CNR4 net radiometer measures the spectral band 0.305 <wavelength ($\lambda$) <2.8 $\mu$m for shortwave radiation and 5< $\lambda$ <50 $\mu$m for longwave. The SR50 senses surface height changes and, thus, can indicate the occurrence of ablation or accumulation but not measure it directly. An * indicates measurements that must be gathered with artificial aspiration in the daytime, a § denotes quantities used to drive ISBA-DEB, and a # marks variables used for calibration or validation. Precipitation was measured hourly at Pyramid Research Station.

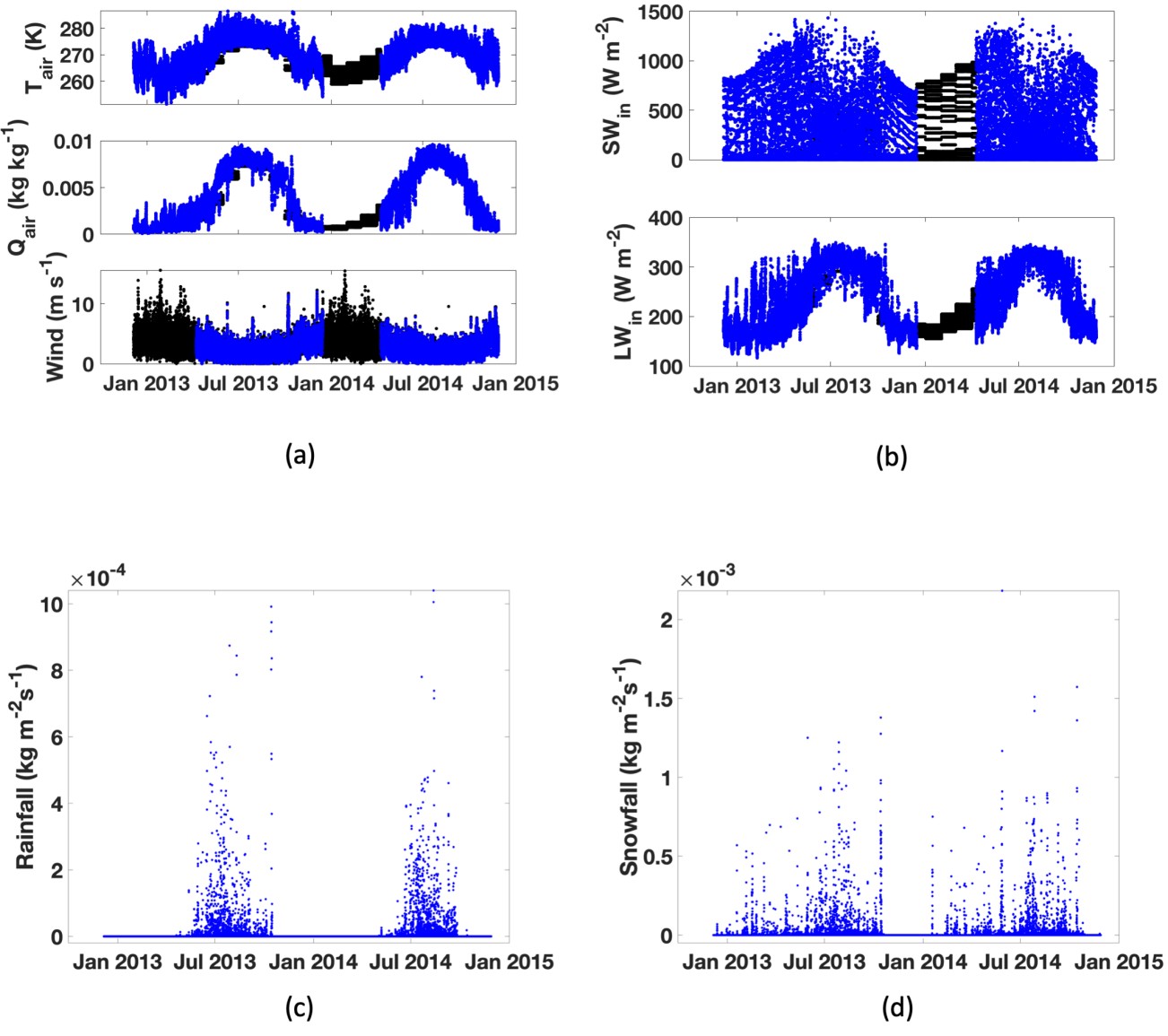

**Figure 2.** (a–b) Continuous half-hourly forcing data in black, with *in situ* data overlaid in blue. The gaps are apparent where the black data points are displayed; Section 2.3 describes the procedure used to assign these values. (c–d) The continuous precipitation dataset from Pyramid (interpolated half-hourly), with phase partitioned by $T_{air}$ at West Changri Nup glacier (top panel of a). Note the different y-axis scales on (c) and (d).

## 2.3 Model Inputs

SURFEX must be forced with temporal and geographic specifications, as well as continuous meteorological variables (complete list at *www.umr-cnrm.fr/surfex/spip.php?article215* and in the Supplementary Material): atmospheric temperature (K), atmospheric humidity (kg kg$^{-1}$), atmospheric pressure (Pa), rainfall rate (kg m$^{-2}$s$^{-1}$), snowfall rate (kg m$^{-2}$s$^{-1}$), wind speed (m s$^{-1}$) and direction (degrees), incoming longwave radiation (W m$^{-2}$), direct shortwave radiation (W m$^{-2}$), diffuse shortwave radiation (W m$^{-2}$), and near surface $CO_2$ concentration (kg m$^{-3}$). As is clear from the § symbols in Table 2, which denotes the measurements used to drive the model, not all of the parameters required to run SURFEX listed above impact the fluxes relevant to integrated models ISBA or ISBA-DEB.

The two-year period used in this study contains data gaps of various lengths affecting different sensors (see Table 2). For example, a battery problem prevented nighttime data readings between April and December 2013 (Figure 5), installation problems made the wind readings questionable for several months, and station tilt compromised the quality of some measurements but not others. Because the forcing file for SURFEX must be continuous, it was necessary to fill such data gaps and periods when data were deemed suspect. See the portion of data plotted with black in Figures 2a and b and the % data gaps in Table 2 for the extent of missing AWS data over the period used in this study.

Missing meteorological values were approximated by the monthly averages of values at the missing timestep during a longer period of data acquisition at the AWS than used for this study: October 2010 – November 2016. Every missing value was filled with the corresponding time step's mean monthly value. Using values specific to timestamps preserved both diurnal and seasonal variability in the gap-filled dataset. This method proved inappropriate for wind speed, whose amplitude and variability could not be conserved with averages. For the whole series, the wind speed data were gap-filled by the wind speed at the same timestamp in a different year of the AWS's operation, randomly selected. When the same timestamp in all years is missing a wind speed, we chose the closest later timestamp with data in any year.

## 3 ISBA-DEB

### 3.1 Model Overview

The ISBA land surface model (Noilhan and Planton, 1989) within the SURFEX community-based open source software platform maintained by Météo-France (Masson et al., 2013) is a physically based scheme that solves both time- and depth-dependent heat and moisture diffusion numerically through mass- and heat-conserving implicit time schemes. It provides a convenient basis for simulating the surface energy balance of a supraglacial debris layer, after making modifications to account for the differences between soil and debris. This work builds on that of Lejeune et al (2013), which used one of the SURFEX snow models, Crocus, to represent dry debris year-round, accounting for snowfall and sub-freezing glacier temperatures during the accumulation season. However, we instead adapted the diffusive version of SURFEX's soil model (ISBA option DIF). As the full details of the ISBA-DIF option for heat and moisture transfer and water phase changes within soil are presented in a series of publications (Boone et al., 1999, 2000; Decharme et al., 2011, 2013), they will be only summarized here to provide

context for the detailed modifications in the supraglacial debris model, ISBA-DEB. By adapting ISBA, we have built a model that not only simulates a supraglacial debris layer's temperature and moisture but also computes glacier melt.

## 3.2 Model Structure

ISBA-DEB computes temperature and moisture in a snow-debris-ice column. Temperature and moisture evolution are calcu-
lated for $10 - 15$ debris layers with user-specified thicknesses. Debris layers are assigned thermal, hydraulic, and physical properties of glacial debris as informed by field measurements on West Changri Nup glacier or, when unknown, by the debris-covered glacier literature. The underlying layers (up to 20 total layers are permitted by the model) approximate a glacier. In ISBA, the glacier layers must be soil, but in ISBA-DEB we assigned them a porosity of 99.9% and specified that they be ice-saturated. Since 99.9% of the volume of these layers is filled with solid ice, glacier layers have an effective porosity of
zero.

Glacier melt water enters the debris at the base, and rain and snowmelt water enter the debris at the surface. Precipitation, wind, air temperature and humidity, and incoming longwave and shortwave radiation measured on West Changri Nup glacier drive the model. We neglect energy carried by precipitation, an assumption supported by other work in the Himalaya (Azam et al., 2014) and on the nearby Tibetan Plateau (Huintjes et al., 2015). A discussion of the forcing variables can be found
in Section 2.3. Figure 3 schematically shows the configuration of the domain and summarizes fluxes and processes in the 1-dimensional ISBA-DEB.

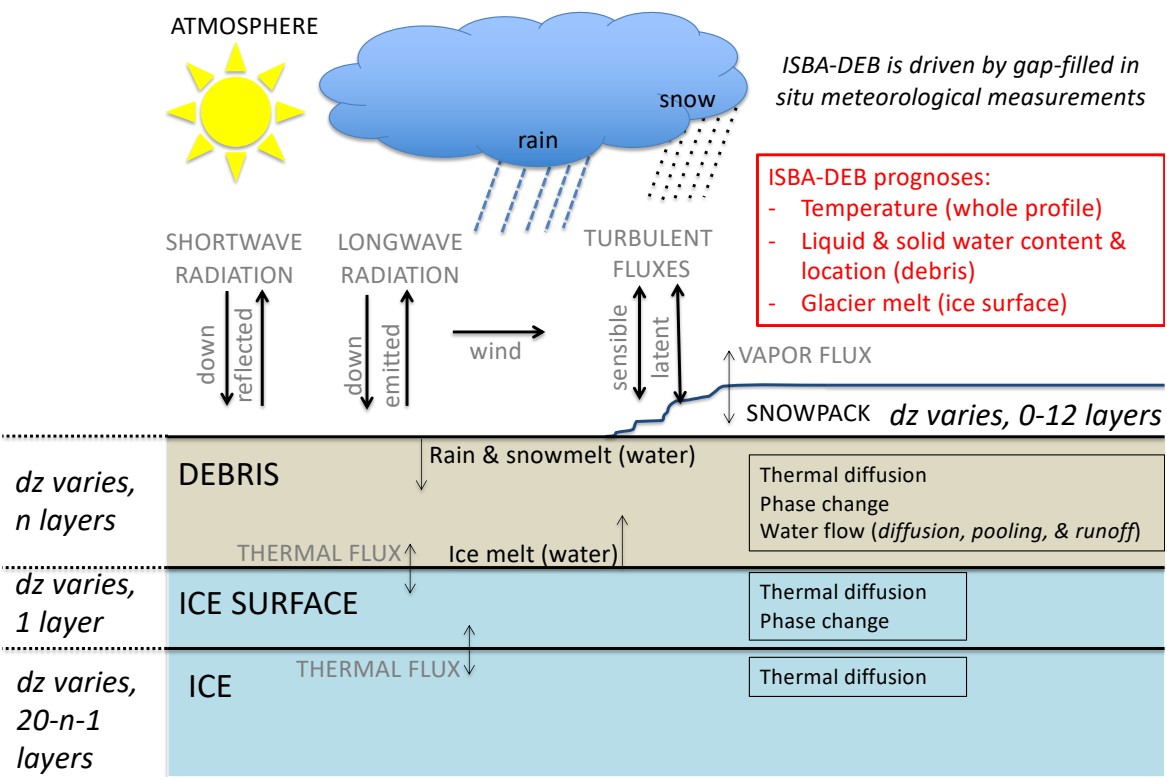

**Figure 3.** General scheme of ISBA-DEB with fluxes and physical processes. Note that ISBA-DEB is a point-scale model but that the schematic is shown in 2D for interpretability. Precipitation comprises part of the system's mass flux and affects debris surface properties; however, the heat it carries is not included in the surface energy budget. This schematic shows the user options for ISBA-DEB. In this study, we use 13 debris layers of 1 cm, an ice surface layer of 1 cm, and 6 additional ice layers (total 20 layers) of varying thicknesses reaching to a depth of 60 m.

ISBA-DEB, like ISBA, solves the temperature in all layers of the domain; the temperature profile is then passed to a routine that computes energy fluxes, including evaporation and glacier melt. The volume of glacier melt and the temperature profile, which has been updated with any melt that occurred during the timestep, pass into the hydrology routines that calculate water volume and location in all allowed layers – as well as its phase according to temperature. Given that the measured debris thickness of 12.5 cm is accurate to $\pm 1$ cm, we use 13 1 cm layers of debris in ISBA-DEB. The prognostic state variables are assumed to be located at the midpoint of each layer; accordingly, the uppermost simulated temperature is at 0.5 cm depth in the debris, not the surface. Under the 13 debris layers are 7 layers of ice, with increasing thicknesses. The layer boundaries in the glacier are at 0.16, 0.45, 2.25, 7.00, 20.0, and 30.0 m in depth. Forced with repeated years of 2013 meteorological data, the model reaches steady state after 40 years of spin up; this is given an initial uniform temperature of 268.35 K and an initial uniform liquid soil water index of 0.1 $m^3$ $m^{-3}$. Other initial conditions require a longer spin up. Above the debris is a transient snowpack, represented by a dynamic 0 – 12 layers. The snow scheme used in ISBA-DEB is ISBA-ES (Boone and Etchevers, 2001; Decharme et al., 2016).

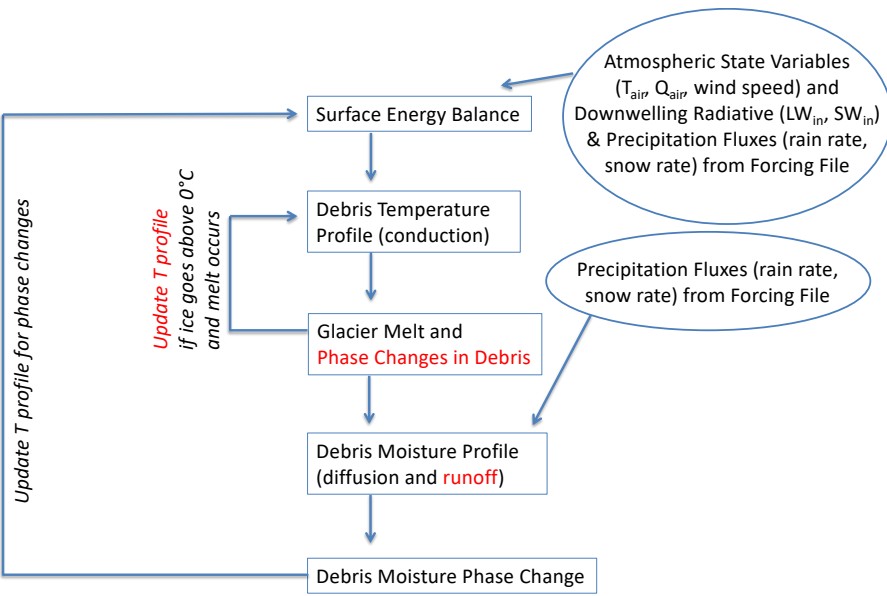

**Figure 4.** Flow of the processes in the ISBA-DEB model. The red text indicates major changes introduced to the ISBA code in the creation of ISBA-DEB. Table A1 contains physical constants and model parameters used for the runs on West Changri Nup glacier.

### 3.3 Physical Processes

### 3.3.1 Heat Diffusion

The ISBA scheme assumes that heat flow along the thermal gradient is the dominant first-order process and neglects other heat transfer processes such as advection within the soil. Such an assumption is common to land surface models currently used in operational numerical weather prediction and general circulation model applications. Heat capacity ($c$) and thermal conductivity ($K$) are weighted averages of the respective volumetric proportions of air, rock, water, and ice (note that the latter is a difference from ISBA).

ISBA-DEB updates the temperature profile for the entire column each timestep using the heat equation in 1-dimension:

$$c_g \frac{\partial T_g}{\partial t} = \frac{\partial}{\partial z} \left[ K \frac{\partial T_g}{\partial z} \right] \tag{1}$$

where $K$ is thermal conductivity (W m$^{-1}$K$^{-1}$), $c_g$ (J m$^{-3}$K$^{-1}$) volumetric ground specific heat capacity, $z$ depth (m), and $T_g$ ground temperature (K). Temperature in debris layers evolves not only by conductive heat transfer but also by latent heat from phase changes between water and ice in the debris ($\Phi$, J m$^{-3}$s$^{-1}$) that is added to the right-hand side of equation 1, giving equation 2. These phase changes are calculated subsequently to the heat transfer routine; the temperature profile is updated accordingly as an adjustment at the end of the timestep (Figure 4).

$$c_g \frac{\partial T_g}{\partial t} = \frac{\partial G}{\partial z} + \Phi \tag{2}$$

In equation 2, conduction flux G represents the term in brackets on the right-hand side of equation 1. A zero flux at depth provides the Neumann lower boundary condition, and surface flux from the energy balance provides the upper boundary condition. Shortwave radiation, longwave radiation, and turbulent fluxes together comprise the surface energy balance. Daily average energy balance components are shown for both winter and summer in Figure S2.

ISBA-DEB calculates temperature for the snow-debris-ice column continuously. However, since the glacier cannot exceed 0 $^o$C, we introduce a condition for the ice layers that follows an analogous scenario for snow in Boone and Etchevers (2001). Only the top layer of ice contributes to the glacier melt term. Underlying ice layers' temperatures are prevented from exceeding freezing by concentrating all above-freezing energy into the melt of the top ice layer. The top layer is 1 cm thick, far greater than the melt possible in a single 15 minute timestep.

### 3.3.2 Glacier Melt

If the top layer of ice exceeds freezing, melt is computed and temperature reset to 0 $^o$C. Sub-surface ice temperatures (i.e. layers 14 – 20) are subsequently recalculated with this 0 $^o$C boundary condition, precluding melt from occurring in the sub-surface layers. Energy is conserved, and the amount of water melted in the top layer of the glacier in each timestep is added to the overlying debris and tracked for a cumulative annual ablation to compare with field measurements. The melting layer is implicitly refilled at the end of the timestep such that the 1 cm thick top layer begins every model iteration at full ice saturation.

### 3.3.3 Moisture Inputs and Diffusion

Water entering the debris from glacier melt and precipitation moves with a vertical flow rate F (m s$^{-1}$) and acts as a source or sink of latent heat $\Phi$ (J m$^{-3}$ s$^{-1}$) from changes phase as a function of temperature. Mass leaves the system through latent heat mass fluxes and runoff ($R$). The amounts of liquid water ($w_l$) and ice ($w_i$), respectively, are given by

$$\frac{\partial w_l}{\partial t} = -\frac{\partial F}{\partial z} - \frac{\Phi}{L_m \rho_w} - \frac{S_l}{\rho_w} - \frac{R}{\rho_w} \qquad (w_{min} \leq w_l \leq w_{sat} - w_i) \tag{3}$$

$$\frac{\partial w_i}{\partial t} = \frac{\Phi}{L_m \rho_w} - \frac{S_i}{\rho_w} \qquad (0 \leq w_i \leq w_{sat} - w_{min}) \tag{4}$$

where $L_m$ is the latent heat of fusion (J kg$^{-1}$), $\rho_w$ is the density of water, and $w_{sat}$ is the water concentration at saturation. $S_l$ and $S_i$ are the external latent heat source/sink terms (kg m$^{-2}$s$^{-1}$) for water and ice, i.e. evaporation for water and sublimation for ice. Values of important physical constants and West Changri Nup glacier-specific parameters are listed in Table A1. A minimum water content $w_{min} = 0.0001$ is retained for numerical stability; $w_{min}$ in ISBA (0.001) was decreased by an order of magnitude in ISBA-DEB given the importance of the exact water content in heat and moisture diffusion calculations (Decharme et al., 2011; LeMoigne, 2018). The change in the liquid moisture in debris is, then, the sum of vertical flow, phase change, inflow and/or evaporation, and runoff; that of ice is the phase change less sublimation.

Vertical soil water flux is given by the Richards' equation and an additive term to account for water vapor. The Richards' equation is an expression derived from Darcy's Law that represents water diffusion arising from pressure gradients in partially saturated media.

$$\frac{\partial w_l}{\partial t} = \frac{\partial}{\partial z}\left[k(w_l)\left(\frac{\partial \psi}{\partial z} + 1\right)\right] \tag{5}$$

Here, $k$ (m s$^{-1}$) is hydraulic conductivity and $\psi$ (m) is soil matric potential, the potential energy attributed to the adhesion of water to soil grains. There have been no observations of ice growth at the surface in subfreezing temperatures on West Changri Nup glacier (as on Mullins glacier, Antarctica by Kowalewski et al., 2011), suggesting that vapor is not a dominant transport mechanism at sub-freezing temperatures. We assume this to be true at temperatures above 0 $^o$C, also, and follow the vapor parameterization of ISBA, where vapor transport is not explicitly modeled on the basis of its being small compared to heat transfer along the thermal gradient and mass flow governed by Darcy's law (i.e. on the basis of scaling arguments). It should be noted that ISBA, in treating vapor transport solely as diffusive, does include an additive term for vapor conductivity. Vertical soil water flux is

$$F = -k\frac{\partial}{\partial z}(\psi + z) - \frac{D_{v\psi}}{\rho_w}\frac{\partial \psi}{\partial z} \tag{6}$$

(Boone et al., 2000), where $D_{v\psi}$ is the isothermal vapor conductivity (kg m$^{-2}$s$^{-1}$) as in Braud et al. (1993).

The Richards' equation (equation 5) includes both diffusion and drainage terms. Observations suggest that moisture transport in glacier debris is neither completely reservoir-like (as parameterized in Collier et al., 2014) nor fully governed by Darcy's

Law (as in the original ISBA for soil) but rather some of both simultaneously. A number of studies (e.g. Collier et al., 2014; Nicholson and Benn, 2012) mention a saturated basal layer of debris, and Rounce and McKinney (2014) discuss deeper, wet debris overlain by dry debris; our own field observations are consistent. The concentration of wetness at the debris base is due both to the fact that debris coarsens upward (Reid and Brock, 2010) and to the permeability of the overlying debris (precipitation quickly moves through the debris until it reaches the impermeable ice surface). By solving the Richards' equation and using an appropriate hydraulic conductivity (Table A1), ISBA-DEB simulates both diffusion and pooling.

Moisture changes phase as a function of available mass and energy (Boone et al., 2000; Giard and Bazile, 2000). As soil freezes, ice is assumed to become part of the soil matrix such that ice lowers debris porosity and enhances the matric potential and vertical upward suction of water.

When there is ice in the debris, equation 6 is rewritten

$$F = -\kappa \frac{\partial \psi}{\partial z} - k \tag{7}$$

where $\kappa = \wp(k + \frac{D_{v\psi}}{\rho_w})$ and $\wp = 10^{-\alpha_\wp} w_i/w$ (Boone et al., 2000). $\wp$ is termed the "ice impedance coefficient," which inhibits upward movement of water towards the freezing front, and $\alpha_\wp$ is the "ice impedance factor," equal to 6 in ISBA (Johnsson and Lundin, 1991) and ISBA-DEB. The form of equation 7 emphasizes that there is a drainage term $k$ and diffusion along a potential $\kappa$ which includes isothermal vapor pressure.

The values of matric potential (m) at saturation, hydraulic conductivity (m s$^{-1}$) at saturation, and shape parameter (dimensionless) of the soil-water retention curve ($\psi_{sat}$, $k_{sat}$, and $b$, respectively) are typically calculated according to Noilhan et al. (1995)'s continuous pedotransfer functions (PTFs), which compute key hydraulic parameters based upon soil composition. For PTF equations, see Appendix C1 of Decharme et al. (2011). Power curves of Brooks and Corey (1966) relate matric potential, hydraulic conductivity, and volumetric liquid water content to the variables computed by PTFs. Values used in our simulations are listed in Table A1.

Instead of using a PTF to calculate $k_{sat}$, ISBA-DEB adopts gravel's $k_{sat}$ value (0.03 m s$^{-1}$, Domenico et al., 1998) throughout the debris except for at the bottommost layer, where $k_{sat} = 0$ m s$^{-1}$. This supplies a flux of 0 for the lower boundary condition, while rainfall and snowmelt provide the upper boundary condition. Equation 3 is solved with a Crank-Nicolson implicit time scheme.

### 3.3.4 Water Runoff

The pebble to gravel-sized grains comprising the debris cannot hold liquid water long-term, and water runs off with a slope-dependent timescale (Zuo and Oerlemans, 1996; Reijmer and Hock, 2008). The timescale is a linear function of glacier surface slope, with values of 1 h$^{-1}$ for $0^o$ and 0 h$^{-1}$ for $90^o$ (Collier et al., 2014) at the surface and an increasing value with depth. Runoff (kg m$^{-2}$s$^{-1}$) can be expressed as

$$R = \frac{w_{l,j} \rho_w \Delta z_j}{\tau_j} \left( \frac{\theta}{90} \right) \tag{8}$$

where $\theta$ is glacier surface slope, measured from horizontal, and $z$ is the thickness (m) of each layer $j$. Runoff timescale $\tau_j$ must be $\leq dt$.

$$\tau_j = \tau_{min} + (\tau_{max} - \tau_{min}) \left[ \frac{exp(\tau_\alpha \frac{z_j}{H}))}{exp(\tau_\alpha)} \right] \tag{9}$$

$\tau_\alpha$ is a tunable shape parameter defining the runoff timescale from its minimum value at the surface (1 hr, Collier et al.,
2014) to its maximum value (also tuned) at the base of the debris, depth $H$ (m). $\tau_\alpha$ controls the distribution of moisture, with larger values leading to a concentration of water at the debris-ice boundary and smaller values leading to a more even distribution. Increasing values of $\tau_\alpha$ show steeper curves, with an increasing number of sub-surface layers having the same moisture content as the surface (Figure S1). All values considered give an increase in water with depth, which is to be expected with the combination of gravity and the fact that debris clasts get finer (with a greater ability to retain water, see Section 1)
with depth.

This parameterization is necessarily simple in the absence of field measurements but corroborated by gravel's high hydraulic conductivity (Domenico et al., 1998) and the observed changes in debris' grain size distribution with depth. Debris grains tend to be smaller in size at the ice surface than at the top of the debris layer, thereby imparting more of a damming effect on entrained water lower in the debris column. The timescale of sand draining is on the order of a day or two (Blum et al., 2018),
indicating an approximate magnitude to inform $\tau_{max}$ tuning tests. Further, debris permeability field tests show that after 10 seconds, $\sim 95\%$ of a 100 mL volume of water poured into gravel and cobbles drains. However, for fine particulates sampled at the ice interface ($< 5$ mm in diameter), only $\sim 20\%$ of the water drains in the same amount of time.

Since it takes a saturated sandy soil $24 - 48$ hours to drain to its field capacity, 48 hours for $\tau_{max}$ is consistent with measurements of the kinds of particles at the base of a debris layer. A shape factor of 30 is consistent with observations of wetted
debris right at the debris-ice interface (Nakawo and Young, 1981; Conway and Rasmussen, 2000; Nicholson and Benn, 2012). A $\tau_\alpha > 30$ does not change the shape of the runoff timescale (green curve in Figure S1) markedly, nor does it improve the RMSE significantly.

Energy and water budgets in ISBA-DEB are the same as those in ISBA, with the exception of an additional term for glacier melt ($M_{ice}$). Both budgets close, and details are presented in the Supplementary Material.

**4   Tuning**

Of the December 2012 – November 2014 series used in this study, we used 2014 debris temperatures to tune parameters and both seasons of ablation to assess the impact of moisture inclusion in ISBA-DEB. We compared simulated debris temperatures

with measured ones from April 9, 2014 – October 23, 2014 (reason displayed in Figure 5), using an RMSE calculation to capture the magnitude of temperature. We tested five runoff timescale shape factors ($\tau_\alpha$, Figure S1) and maximum runoff timescale ($\tau_{max}$) values of 3, 6, 18, 24, 48, 72, and 96 hours. The RMSE metric suggested a shallow minimum for $\tau_\alpha =30$ and negligible differences for different $\tau_{max}$ values. We used $\tau_\alpha =30$ and $\tau_{max} =48$ for our modeling work, despite the shallow minima, because they are highly plausible values.

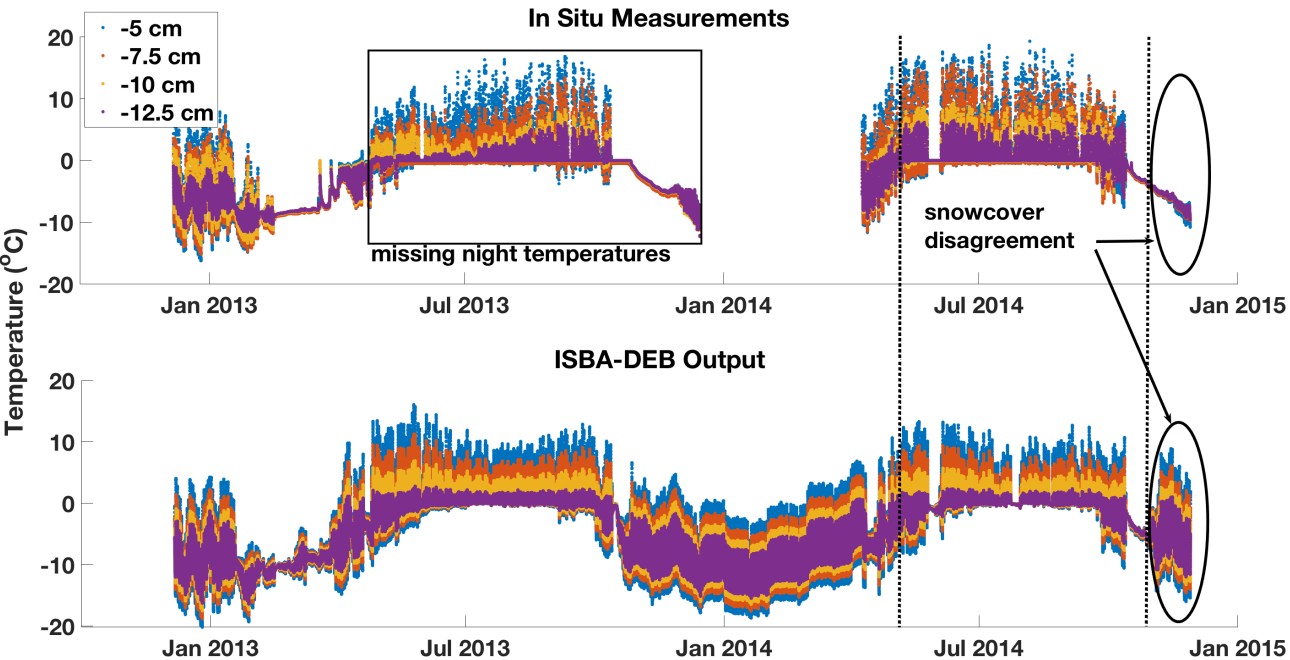

**Figure 5.** Measured (upper panel) and modeled (lower panel) debris temperatures at depths of 5 cm, 7.5 cm, 10 cm, and 12.5 cm in a 12.5 cm thick debris layer, displayed to show that the period between the vertical dashed lines, April 9, 2014 – October 23, 2014, was most informative for comparison in tuning. There was a battery problem causing no nighttime temperature recordings April – December 2013 (period indicated by the black box), and the clear temperature disagreement in late 2014 results from a problem in the meteorological forcing file for ISBA-DEB (having insufficient snowfall to produce the observed persistent snowcover). The model runs in the lower panel were carried out with $\tau_\alpha = 30$ and $\tau_{max} = 48$ hr, and closer look at modeled and measured temperatures during the period of comparison is given in Figure S3.

# 5 Results and Discussion

In this section, we present the results (and describe the behaviour) of model simulations for nearly two years of meteorological forcing, describe key physical processes related to the presence of debris, and show results from a series of sensitivity tests related to parameter uncertainties.

## 5.1 Model Simulation Characteristics

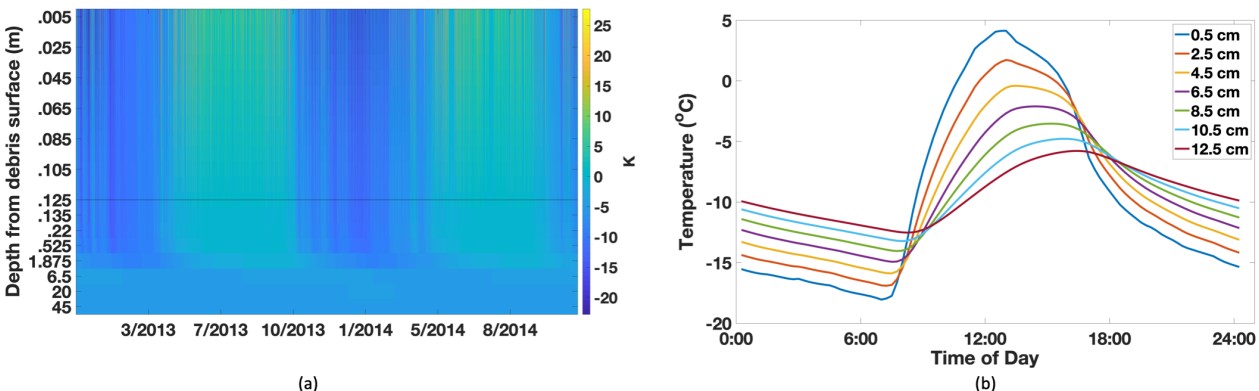

(a)

(b)

**Figure 6.** (a) Temperatures in the debris and underlying glacier throughout the period used for this study. The black line indicates the debris-ice interface. The depth scales for debris and ice differ; debris thickness is 0.125 m whereas the ice extends to 60 m (note the non-linear y-scale in the ice). Y-labels correspond to the mid-layer depth of the 20 discrete layers used in ISBA-DEB. The temperatures simulated throughout the whole 60 m column over the entire forcing period show phase lag and attenuation with depth, characteristics that are more clearly seen in (b), which shows the temperature of various debris depths during an arbitrary day (20 February 2014).

During the model simulations, glacier melt, snowmelt, and rain enter the debris base or surface. The moisture in each layer evolves with time, and the phase of the moisture changes as a function of temperature. Figure 7 illustrates debris water input in the top (surface) and bottom (debris-ice interface) panels.

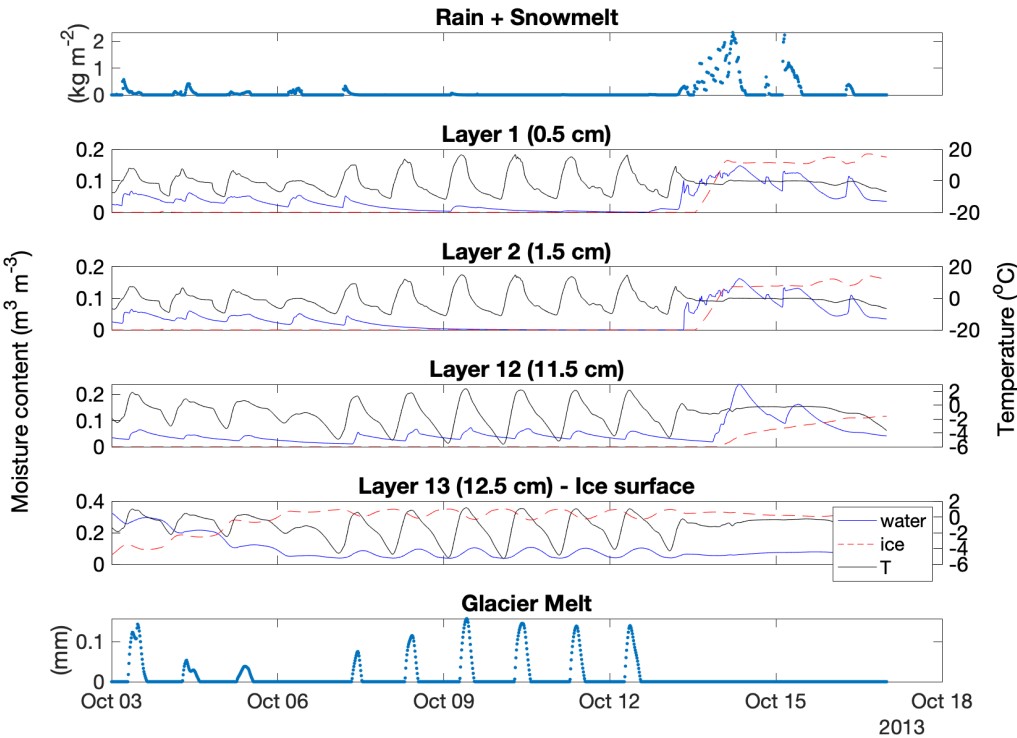

**Figure 7.** Time-evolving water content, ice content, and temperature of the top two and bottom two layers of debris, shown during a 2-week period when water enters the debris both at its surface (from rain and snowmelt, top panel) and at its base (from glacier melt, bottom panel). The phase of the moisture changes as a function of temperature. Note that the moisture and temperature y-scales vary between the layers.

Its middle four panels show how the liquid and solid moisture contents change with temperature in the top two and bottom two layers of debris (i.e. layers 1, 2, 12, and 13). ISBA-DEB simulates temperature evolution throughout the entire debris-glacier model domain (Figure 6a); the domain is 60 m total, including the 13 debris layers, each 1 cm thick. Output shows temperature amplitude attenuation and phase lag with depth (clearly seen in Figure 6b). The above-freezing temperatures propagating into the ice cause melt (Figure 7). Cumulative melt at each time step (Figure 8, blue dots) gives the total melt (Figure 8, red line).

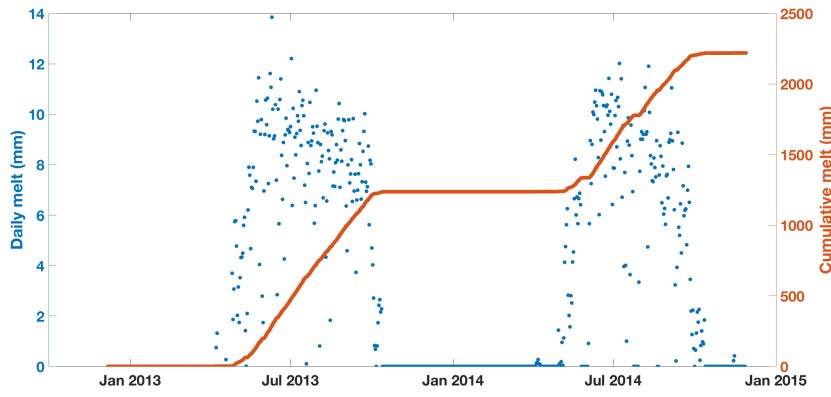

**Figure 8.** Daily totals of modeled glacier melt (blue dots), overlaid by modeled cumulative melt (red line) for the entire period December 2012 – November 2014 simulated by ISBA-DEB. For a visual depiction of how and when melt occurs in ISBA-DEB, see Figure 7.

As the debris' moisture content and phase vary, its thermal conductivity and heat capacity evolve accordingly (Figure 9; extreme values are listed in Table 1). Layers 1 – 12 look similar because the moisture is concentrated in layer 13, which is just above the ice-debris interface. (Note that the AWS on West Changri Nup has a slope of $5^o$ and that a flatter debris layer may hold more water, with moisture concentrated in more of the lower layers than solely layer 13.) During the summer, the glacier is

melting, and the bottommost layer of debris is almost always saturated with liquid water, such that it has a thermal conductivity (K) of $\sim$1.16 W m$^{-1}$K$^{-1}$ (Table 1, Figures S5a, c). The thermal conductivity of layer 13 changes little throughout the day; layer 1 shows slight variation in conductivity because its water content experiences variation via condensation and evaporation. For conductivity in the summer (Figure S5c), a higher value means more water content while a lower value indicates dryness. The summer monsoon season (JJAS) mean diurnal patterns in conductivity (Figure S5c) are similar to those in specific heat

capacity (Figure S5d) because both are functions of water content, and both thermal conductivity and heat capacity have higher values for the water-saturated debris in layer 13 than the drier debris in the overlying layers – and vary little throughout the day. For layer 13 at the ice-debris interface (12.5 cm), conductivity is greatest at the transition into winter, when the water filling the pore spaces freezes (Figure S5a). Conductivity is greater for ice-saturated debris than for water-saturated debris, which is still greater than for dry debris. Heat capacity is greatest for water-saturated debris and less for ice-saturated debris (still less

for dry). As expected, heat capacity is greater in the summer than winter in the bottommost debris layer (Figure S5b). The temporal and spatial evolution of these parameters throughout the debris column as a function of water and ice contents is a strength of ISBA-DEB.

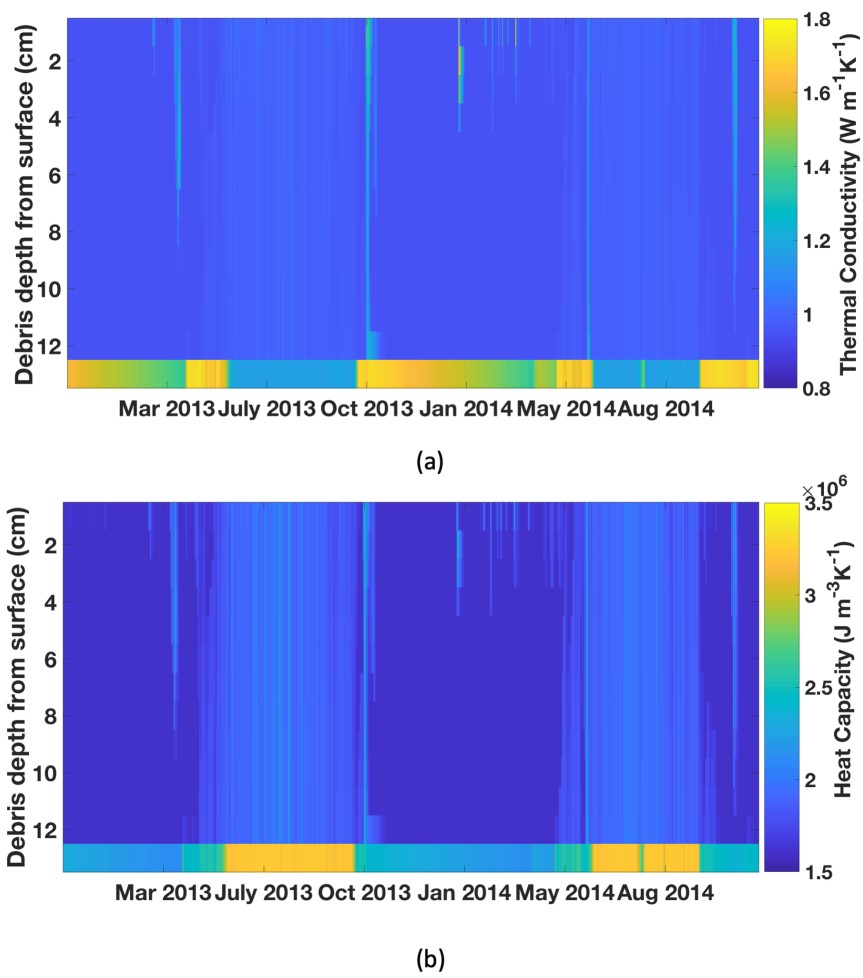

**Figure 9.** Temporal evolution of (a) thermal conductivity and (b) volumetric heat capacity according to debris moisture amount, phase, and gradient. Supplementary Figure S5 gives a different perspective on the evolution: temporal changes in the the debris surface, middle, and base layers, as well as the 2013 monsoon season (JJAS) diurnal averages for each quantity.

## 5.2 Wet versus Dry Debris

We ran an experiment to contrast the sub-debris melt under totally dry, partially saturated, and fully saturated debris layers forced with the same meteorological conditions (Figure 2) measured on West Changri Nup glacier between December 2012 and November 2014. The "partially saturated" scenario uses parameters listed in Table A1. In order to achieve fully dry debris, all rain and melted snow was assumed to run off immediately, although solid precipitation (snow) could persist and sublimate. Table 3 shows the three computed point mass balance values (not accounting for runoff) compared to available measurements. A bamboo stake, which carries an uncertainty of $\pm 200$ mm w.e. (Vincent et al., 2016) and SR50-detected surface height changes, using a snow density of 200 kg m$^{-3}$, supply the measurements. In 2014 the stake broke, and SR50 was operational from only 09/04/2014 to 19/07/2014.

| Mass Balance Components (mm w.e.) | Completely Dry Scenario (mm w.e.) | Partially Saturated Scenario (mm w.e.) | Fully Saturated Scenario (mm w.e.) | Measurements (mm w.e.) |
|---|---|---|---|---|
| 5/12/2012 to 02/12/2013 | | | | |
| Cumulated solid precipitation | 289 | 289 | 289 | |
| Melt | 1241 | 1237 | 771 | |
| Sublimation | 40 | 40 | 220 | |
| Evaporation | 0 | 81 | 549 | |
| Ablation | 1281 | 1359 | 1540 | |
| Point mass balance | -992 | -1069 | -1251 | -1080 (at SR50) -753 (at stake) |
| 02/12/2013 to 28/11/2014 | | | | |
| Cumulated solid precipitation | 368 | 368 | 368 | |
| Melt | 975 | 983 | 605 | |
| Sublimation | 15 | 15 | 441 | |
| Evaporation | 0 | 66 | 553 | |
| Ablation | 990 | 1064 | 1599 | |
| Point mass balance | -622 | -696 | -1231 | N/A |
| 09/04/2014 to 19/07/2014 | | | | |
| Cumulated solid precipitation | 142 | 142 | 142 | |
| Melt | 500 | 495 | 267 | |
| Sublimation | 8 | 8 | 338 | |
| Evaporation | 0 | 47 | 362 | |
| Ablation | 508 | 550 | 967 | |
| Point mass balance | -366 | -408 | -825 | -760 (at SR50) |

**Table 3.** Mass balance components of three model runs (dry debris, partially saturated debris, and fully saturated debris) compared with the available measured point mass balance from 5/12/2012 to 19/07/2014. The observation-driven model behavior and output presented in previous figures is from what is termed the "partially saturated scenario" here. Note that runoff has not been taken into account for this comparison.

Important characteristics of Table 3 include the dry debris' zero evaporation but sublimation commensurate with that of the partially saturated debris; this results from the assumption that rain and melt run off instantaneously while SURFEX still models a snow cover that can sublimate. The sublimation computed in the fully saturated scenario is a sum of snow sublimation and debris water sublimation that occurs when a snow cover is absent in sub-freezing temperatures.

The glacier under completely dry debris melts significantly more than the glacier situated under fully saturated debris in all three periods (Table 3). The glacier under partially saturated debris gives a simulated melt close to that under dry debris. In 2013, the SR50 data are not in agreement with the single ablation stake only a few meters away because the spatial variability of point mass balance can be very high over a few meters of debris cover (Vincent et al., 2016), due to differences in debris thicknesses, properties, and water content. Overall, there is a reasonable agreement between measured and modeled point mass
balances, lending confidence to the simulations.

Sub-debris melt is a function of the debris thickness, which is the same for all three cases, and the thermal diffusivity of the debris ($K/c_g$ in equation 1), which differs for all three as a function of the amount, phase, and location of moisture. Completely water-saturated debris has a thermal diffusivity that is less than half of the diffusivity for completely ice-saturated debris. Dry debris' diffusivity falls nearly midway between the two (Table 1). The share of water and ice in the interstitial
spaces of the partially (Figure 10a) and fully (Figure 10b) saturated debris differs significantly in amount and distribution. Ice-saturated debris conducts heat much more efficiently than water-saturated debris does; however, glacier melt happens only when the glacier surface exceeds 0 $^o$C, and efficiently conducting, ice-laden debris overlying a melting glacier is a physical impossibility. The fully saturated debris conducts heat that leads to melt with an efficiency of $\kappa = 3.572 \times 10^{-7}$ m$^2$ s$^{-1}$, while the dry debris has a diffusivity of $5.867 \times 10^{-7}$ m$^2$ s$^{-1}$; hence, more melt occurs below the dry debris in ISBA-DEB.

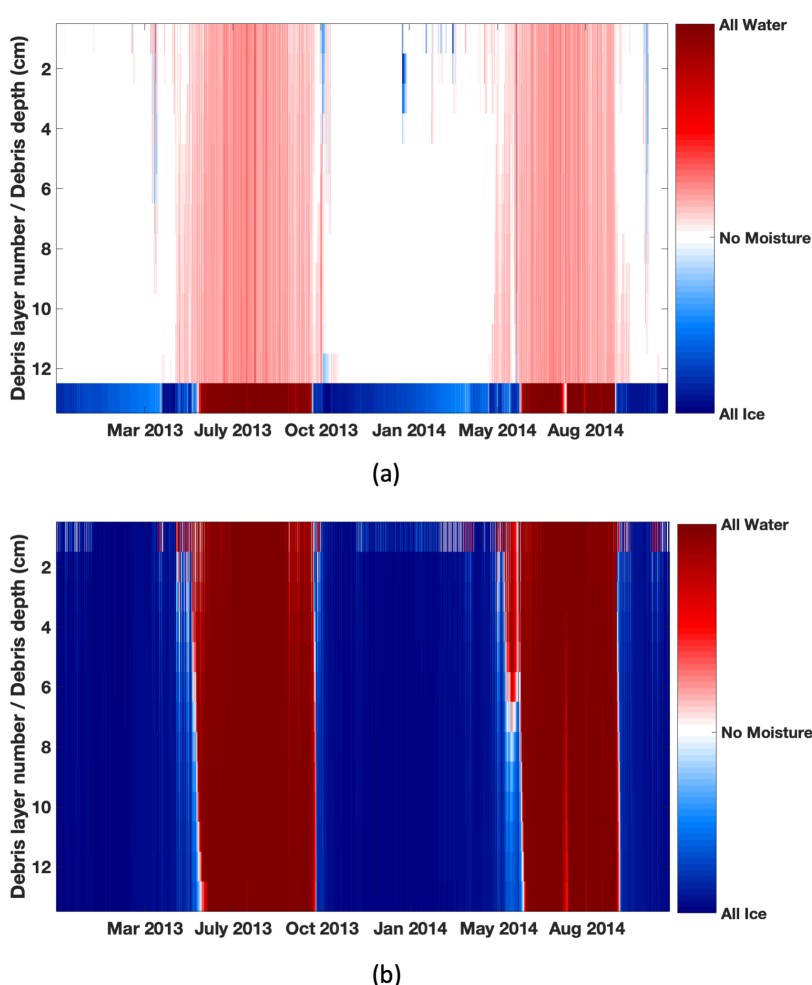

**Figure 10.** Debris layer moisture by phase in (a) the partially saturated ISBA-DEB and (b) the fully saturated ISBA-DEB scenarios. As shown in (b), the fully saturated debris is water-filled (having a lower thermal diffusivity than dry debris) during the summers when debris surface temperatures lead to glacier melt. The difference in moisture in the debris surface layer accounts for the different surface latent heat fluxes (Figure 11). We refer the interested reader to the 2013 JJAS mean diurnal water and ice contents (Figure S4).

As expected, the surface latent heat flux is much greater over the saturated debris, and the latent heat flux due to phase changes within the debris is also greatest for the saturated debris (Figure 11). In the scenarios compared here, incoming shortwave and longwave components are the same. Outgoing radiative fluxes will vary mainly based on surface albedo and surface temperature, respectively (they are both modulated by the absence or presence of snow cover). Latent heat varies most

5   between the scenarios and has the greatest impact on surface energy balance and, thus, its residual, the conductive heat flux into the debris and ultimately transferred to the underlying glacier. (For completeness, note that sensible heat flux is larger for a dry surface where less latent heat transfer is taking place than for a wet surface but is comparatively smaller in magnitude.) Even in the fully saturated debris, the latent heat from freeze-thaw of the debris is two orders of magnitude less than the latent heat from evaporation and sublimation. The significant energy used for evaporation and sublimation leaves comparatively little

10   energy for heat conduction through the debris-ice interface when the debris layer is fully saturated and water-filled (or, in winter, ice-filled); indeed, when latent heat is significant, the surface doesn't heat up, and the temperature gradient controlling conduction is weak. Therefore, not only does a wet debris layer transfer heat less efficiently from its surface to its base than dry debris because of a decreased thermal diffusivity, but also it has less energy to transfer in the first place because of the other energy fluxes (mainly the surface latent heat) associated with the scenario.

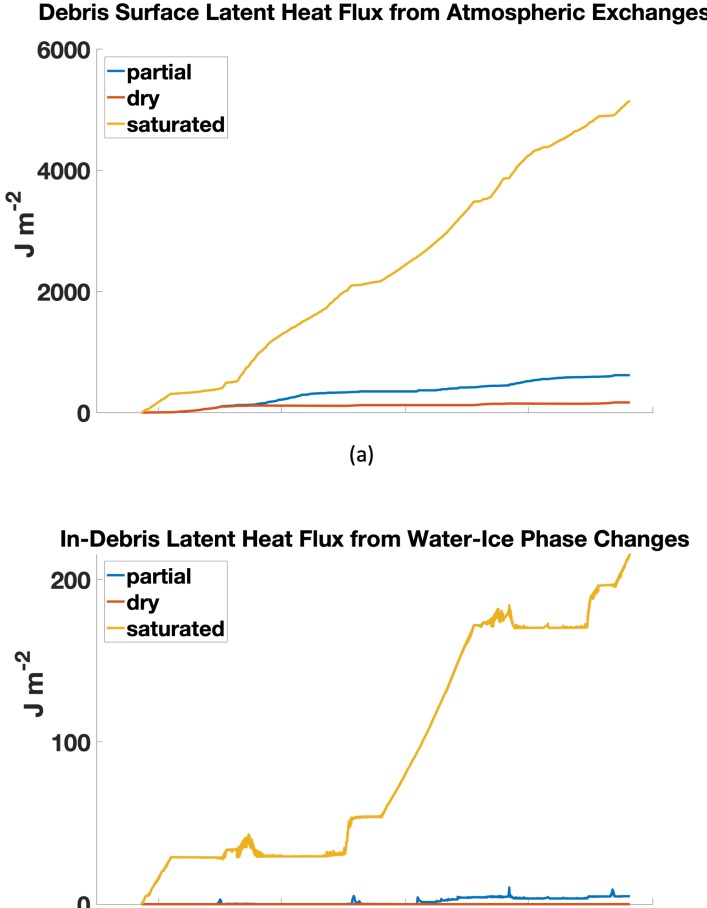

**Figure 11.** Cumulative heat fluxes in the three scenarios (dry debris, partially saturated debris, and fully saturated debris). The surface latent heat flux exchanged with the atmosphere (a) indicates how much energy is removed from the system at the debris surface, while the latent heat from phase changes within the debris (b) shows how much energy is removed from the system in the sub-surface of the debris layer. Greater latent heat fluxes are balanced with a lower conductive heat flux through the debris into the underlying glacier.

Overall, our results show that including moisture in supraglacial debris with ISBA-DEB over 2012 – 2014 on West Changri Nup glacier does not significantly decrease sub-debris glacier melt as compared to a dry debris layer – until the debris is fully saturated and the top layer holds a significant amount of moisture. In general with ISBA-DEB, any change in melt under partially saturated debris is determined by the distribution and amount of water in the debris because water decreases the debris layer's bulk thermal diffusivity and causes it to lose energy, which otherwise might be used for ice melt, to latent heat fluxes. The amount of melt is highly sensitive to the runoff parameterization, the assumptions made in ISBA-DEB, and the meteorological forcing. The partially saturated debris is predominantly dry, with the exception of the lowermost layer (Figure 10a, matching observations described in Section 3.3.3). With different runoff parameters, a flatter slope, and/or more water entering the debris from precipitation, the partially saturated debris scenario could yield an annual ablation closer to the value for saturated debris in Table 3. ISBA-DEB follows ISBA's calculation of atmospheric latent heat exchange from the top layer only. Introducing an atmospheric latent heat flux within the debris similar to that at the "saturated horizon" of Collier et al. (2014) or a wind flow parameterization as in Evatt et al. (2015) would give lower glacier melt underlying a partially saturated debris layer.

## 5.3 Sensitivity Tests

We performed sensitivity tests on the six parameters listed in Table 4 to explore and quantify uncertainty associated with parameterizations in ISBA-DEB. In most cases, the tested ranges were informed by literature. In the case of albedo, which has been found to vary up to 0.6 on debris-covered glaciers in the Everest region (Kayastha et al., 2000), we tested values ranging from 0.1 – 0.5; Kayastha et al. (2000) claimed that most albedo values fall in the 0.2 – 0.4 range, while Nicholson and Benn (2012) showed that 62% of their measurements fell between 0.1 and 0.3. A mid-day mean of the ratio of reflected to incoming shortwave radiation measured on West Changri Nup glacier gives an albedo of 0.2. Despite the fact that albedo has been measured on West Changri Nup glacier, ISBA-DEB's sensitivity to this parameter is important to assess for future application of ISBA-DEB to other debris covers.

A study on Miage glacier, Italy provided 0.94 W m$^{-1}$K$^{-1}$ as a starting point for thermal conductivity tests (Reid and Brock, 2010), though we varied the conductivity values throughout the range reported in the literature, 0.60 to 1.29 W m$^{-1}$K$^{-1}$ (Rounce et al., 2015). This was a particularly important sensitivity test to perform because, as noted in the caption to Table 1, the thermal values we assumed valid for dry debris on West Changri Nup glacier were "effective" values reported for Miage glacier. As Brock et al. (2010) measured thermal conductivity in the ablation season, the reported K of 0.94 W m$^{-1}$K$^{-1}$ (Reid and Brock, 2010) was likely higher than that of perfectly dry Miage glacier debris since any moisture in the pore spaces would have had K = 0.57 W m$^{-1}$K$^{-1}$ (water) rather than K = 0.024 W m$^{-1}$K$^{-1}$ (air). Additionally, debris on Miage glacier (Italy) may have a dramatically different lithology than the debris on Changri Nup glacier (Nepal). Reports of debris conductivity on Khumbu glacier, adjacent to Changri Nup, include 0.85 W m$^{-1}$K$^{-1}$, 1.28 W m$^{-1}$K$^{-1}$ (Conway and Rasmussen, 2000), and 0.96 W m$^{-1}$K$^{-1}$ (Rounce and McKinney, 2014) and indicate that 0.94 W m$^{-1}$K$^{-1}$ is not inappropriate to apply to West Changri Nup. West Changri Nup's debris is most likely comprised of the sillimanite gneiss that forms its surrounding mountains (Searle et al., 2003). The USGS's "Thermal Properties of Rocks" (Robertson, 1988) gives a thermal conductivity of

| Parameter | Cumulative Ablation (mm) Dec. '12 – Nov. '14 | % Change in Melt (relative to *) | % Change in Parameter Range (from *) |
|---|---|---|---|
| $\alpha = 0.1$ | 2425.10 | 9.25 | -20 |
| $\alpha = 0.2$* | 2219.80 | – | – |
| $\alpha = 0.3$ | 2001.10 | -9.85 | 20 |
| $\alpha = 0.4$ | 1766.60 | -20.42 | 40 |
| $\alpha = 0.5$ | 1510.40 | -31.96 | 60 |
| $K = 0.6\,\mathrm{W\,m^{-1}K^{-1}}$ | 1533.30 | -30.93 | -49 |
| $K = 0.7\,\mathrm{W\,m^{-1}K^{-1}}$ | 1760.80 | -20.68 | -34 |
| $K = 0.8\,\mathrm{W\,m^{-1}K^{-1}}$ | 1966.20 | -11.42 | -20 |
| $K = 0.94\,\mathrm{W\,m^{-1}K^{-1}}$* | 2219.80 | – | – |
| $K = 1.0\,\mathrm{W\,m^{-1}K^{-1}}$ | 2317.20 | 4.39 | 8.6 |
| $K = 1.1\,\mathrm{W\,m^{-1}K^{-1}}$ | 2466.40 | 11.11 | 23 |
| $K = 1.2\,\mathrm{W\,m^{-1}K^{-1}}$ | 2603.10 | 17.27 | 37 |
| $K = 1.3\,\mathrm{W\,m^{-1}K^{-1}}$ | 2730.70 | 23.02 | 51 |
| $z_{o,m} = 0.0035\,\mathrm{m}$ | 3081.90 | 38.84 | -9.4 |
| $z_{o,m} = 0.0063\,\mathrm{m}$ | 2879.50 | 29.72 | -8.8 |
| $z_{o,m} = 0.016\,\mathrm{m}$ | 2575.20 | 16.01 | -6.9 |
| $z_{o,m} = 0.05\,\mathrm{m}$* | 2219.80 | – | – |
| $z_{o,m} = 0.1\,\mathrm{m}$ | 2008.40 | -9.52 | 10 |
| $z_{o,m} = 0.5\,\mathrm{m}$ | 1502.30 | -32.32 | 91 |
| $z_{o,h} = 0.05\,\mathrm{m}$ | 1426.20 | -35.75 | 91 |
| $z_{o,h} = 0.0125\,\mathrm{m}$ | 1945.70 | -12.35 | 15 |
| $z_{o,h} = 0.0071\,\mathrm{m}$ | 2115.30 | -4.71 | 4.2 |
| $z_{o,h} = 0.005\,\mathrm{m}$* | 2219.80 | – | – |
| $z_{o,h} = 0.002\,\mathrm{m}$ | 2479.60 | 11.70 | -6.0 |
| $z_{o,h} = 0.001\,\mathrm{m}$ | 2663.30 | 19.98 | -8.1 |
| $z_{o,h} = 0.0005\,\mathrm{m}$ | 2827.60 | 27.38 | -9.1 |
| $z_{o,h} = 0.0003\,\mathrm{m}$ | 2949.70 | 32.88 | -9.5 |
| $\epsilon = 0.9$ | 2246.90 | 1.22 | -40 |
| $\epsilon = 0.94$* | 2219.80 | – | – |
| $\epsilon = 1$ | 2179.90 | -1.80 | 60 |
| $\theta = 0^{o}$ | 1479.40 | -33.35 | -50 |
| $\theta = 1^{o}$ | 1539.90 | -30.63 | -40 |
| $\theta = 2^{o}$ | 1694.30 | -23.67 | -30 |
| $\theta = 3^{o}$ | 1908.20 | -14.04 | -20 |
| $\theta = 4^{o}$ | 2094.60 | -5.64 | -10 |
| $\theta = 5^{o}$* | 2219.80 | – | – |
| $\theta = 6^{o}$ | 2284.30 | 2.91 | 10 |
| $\theta = 10^{o}$ | 2320.40 | 4.53 | 50 |

**Table 4.** Summary of sensitivity tests performed on ISBA-DEB on albedo ($\alpha$), thermal conductivity (K), roughness lengths for momentum ($z_{o,m}$) and heat ($z_{o,h}$), emissivity ($\epsilon$), and slope ($\theta$). An asterisk indicates values that are used in Section 5.1 and for the partially saturated scenario in Section 5.2. Each parameter was varied while the others were held at their values with an asterisk. These values provide the basis of comparison in columns 3 and 4.

$2\,\mathrm{W\,m^{-1}K^{-1}}$ (see Figures 3 and 16 therein). For debris with 39% porosity and air-filled pore spaces, a weighted-average K for dry debris is $1.2\,\mathrm{W\,m^{-1}K^{-1}}$, which is within our tested range. Thermal conductivity is difficult to measure in the field, and it is not known how transferable the limited available measurements are to other debris covers and conditions. It is also not known whether a weighted average of bedrock and air thermal properties is a valid representation of porous debris. Accordingly, we intended to encompass the true value(s) in the range for which we tested ISBA-DEB's response.

Aerodynamic roughness lengths are used to determine the two exchange coefficients ($C_H$, $C_D$) in the stability correction for the bulk method of calculating turbulent heat fluxes (i.e. fits to the Monin-Obukhov functions, see Noilhan and Mahfouf, 1996). $C_D$ (for momentum) depends on $z_{0,m}$, while $C_H$ (for H and LE) depends on both $z_{0,m}$ and $z_{0,h}$. The surface roughness

length due to momentum, $z_{0,m}$, is the height above a rough surface at which the horizontal wind speed is zero. It varies with time and snowfall, and it is notoriously poorly constrained (Quincey et al., 2017) and difficult to compute consistently with different approaches (Miles et al., 2017). The values of both roughness lengths are inherently difficult to measure and poorly known because they depend on not only the local surface state but also meteorology and surrounding surface features.

Studies that informed our range of tested values were: Inoue and Yoshida (1980) and Takeuchi et al. (2000) for 0.0035 m and 0.0063 m on Khumbu glacier, respectively; Reid and Brock (2010) for 0.016 m on Miage glacier; and Lejeune et al. (2013) for 0.05 m determined through model tuning on West Changri Nup glacier. We test 0.1 m, reasoning that debris' roughness can be approximated by that of rough ice (Smeets and Van den Broeke, 2008). An upper end member, 0.5 m, is taken from Miles et al. (2017)'s value for boulders on Lirung glacier. Their value for gravels (0.005 m) and Quincey et al. (2017)'s recent
measurements at two sites on Khumbu glacier (0.0184 and 0.0243 m) fall within the range of tested values.

The roughness length of heat transfer ($z_{0,h}$) is incorporated into ISBA through the variable $z_{0,m}/z_{0,h}$, which must be $\geq 1$. The smaller this ratio, the larger $z_{0,h}$ and the larger $C_H$ (and turbulent flux). $z_{0,m}/z_{0,h}$ is commonly taken to be $= 10$ (ISBA default, Mascart et al., 1995), but we test a wide range for ISBA-DEB given the uncertainty surrounding the value of this parameter. We test ratio values of 1, 4, 7, 10, 25, 50, 100, and 200.

Emissivity affects net longwave radiation and other surface fluxes through feedbacks; we test the model's response to a wide range of values for this parameter (i.e. $0.9 - 1$). Finally, we test how sensitive model-simulated melt is to the user-specified slope that determines runoff. We test a range from flat to a slope of $10^o$. Figure 12 summarizes cumulative melt over the entire two year period for the extreme parameter values tested, and Supplementary Figure S6 shows cumulative melt for all parameter values in Table 4.

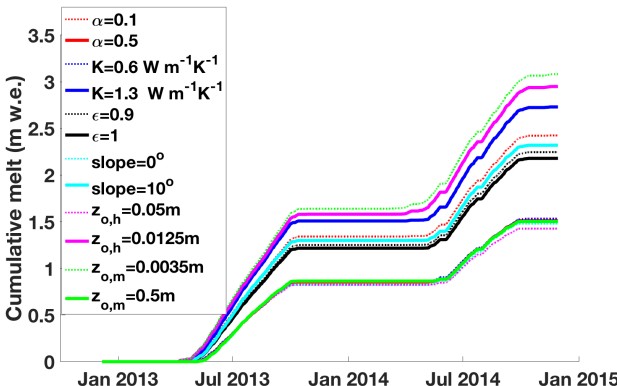

**Figure 12.** Cumulative glacier melt over the December 2012 – November 2014 period under the extreme values for each parameter listed in Table 4. An equivalent figure for each parameter may be found in Figure S6.

As shown in the subplots of Supplementary Figure S6 and associated Table 4, the ISBA-DEB model would give significantly different melt results on glaciers with a much more reflective debris cover (i.e. a lithology with a higher albedo), a much flatter surface, or different $z_{0,m}$ and $z_{0,h}$ values. Given the responsiveness of ISBA-DEB's calculated melt to thermal conductivity, we

elected to compare simulated and measured debris temperatures to glean information about which value of thermal conductivity yields simulated debris temperatures that most closely match measured ones in timing (via $R^2$ of envelope functions) and magnitude (via RMSE). Our tests do suggest an optimal value of 1 W m$^{-1}$K$^{-1}$, which agrees closely with that of Reid and Brock (2010), though further tests over more time periods with available debris temperatures are necessary because neither of these tests yielded deep minima.

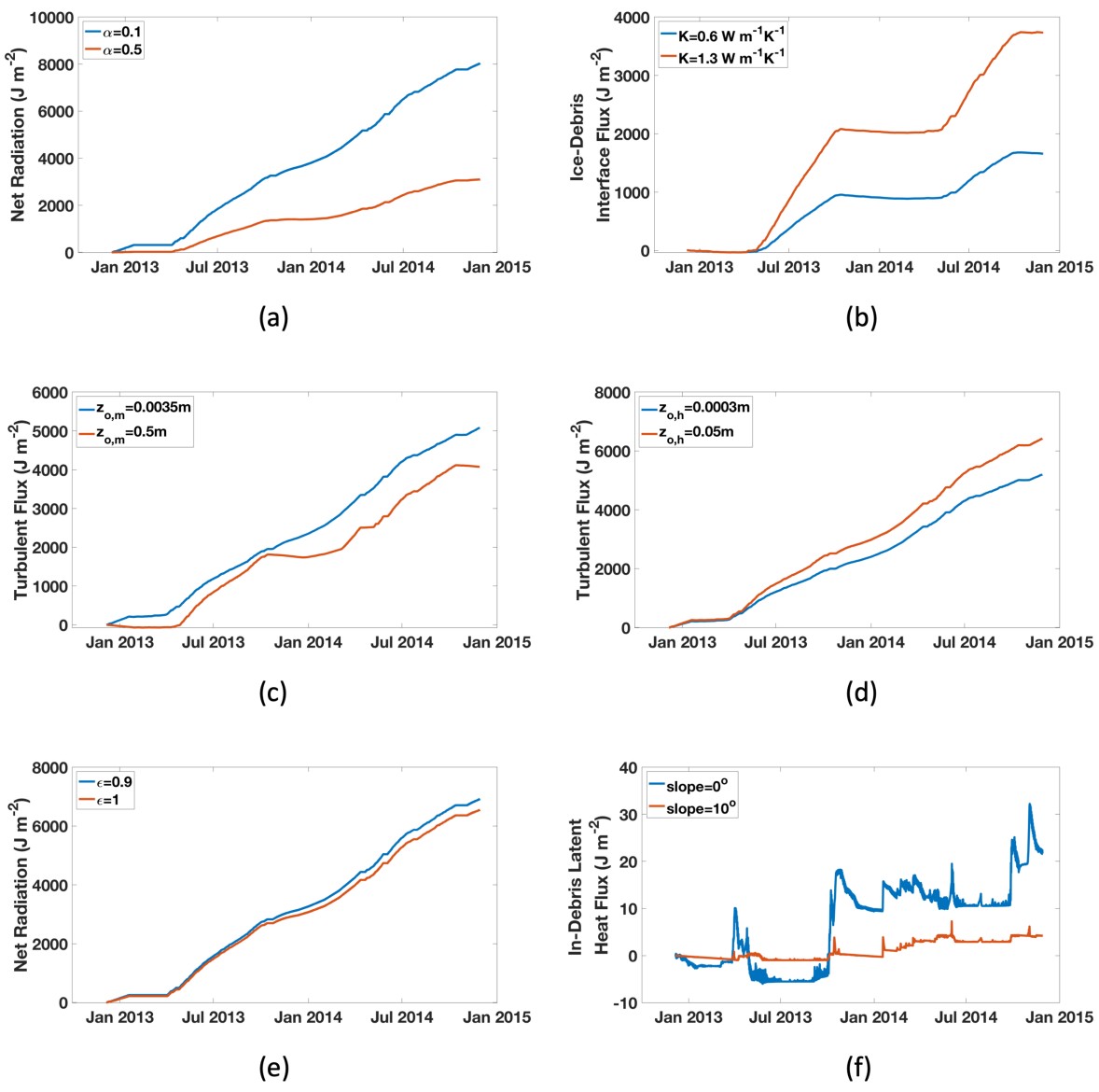

**Figure 13.** Cumulative energy fluxes most impacted by the six parameters perturbed through sensitivity tests, shown with the maximum and minimum tested values. Subfigure locations of Figure S6 correspond.

Figure 13 accompanies Figure S6 in that it shows which energy fluxes have the strongest impact on the variation of the cumulative melt curves (shown in Figure S6). Roughness lengths determine the surface turbulent heat fluxes, and albedo and emissivity affect net radiation. Thus, these four parameters determine how much energy is available to enter the debris. Both slope (which effectively alters the thermal properties of the debris by controlling the residence time of water) and thermal conductivity determine how much energy reaches the ice surface to melt it. (Note that, by modulating water content in the debris, slope has a large effect on the flux of latent heat due to phase changes within the debris.) When water is available at the surface, as with a flatter slope or the fictional "fully saturated," wet scenario in Section 5.2, ISBA-DEB's simulation of sublimation and evaporation removes energy and reduces ice melt. Thermal conductivity, by definition, has the greatest impact on the conductive heat flux, the cumulative value of which varies from 1658 J m$^{-2}$ for K = 0.6 W m$^{-1}$K$^{-1}$ to 3734 W m$^{-2}$ for K = 1.3 W m$^{-1}$K$^{-1}$.

Because roughness lengths are difficult to measure, they are unlikely to be well constrained; nevertheless, they are parameters to which ISBA-DEB is extremely sensitive. Table 4 gives the % change in melt resulting from the parameter change and the % that the parameter value was perturbed relative to the plausible range, which was tested. Because roughness lengths are so poorly known, we tested a large range that was non-linear in its distribution. The high sensitivity to roughness length perturbations (a 9% decrease in the value of roughness length for momentum resulting in a 39% increase in melt, for example) is partly reflective of the size and distribution of the range tested but also underscores the model's sensitivity to these parameters and the importance of focusing future work on them. See Section 5.5 for a discussion of distributed modeling.

Our simulations showed that sensible heat flux (H) is one order of magnitude larger during the monsoon and a factor of seven larger during the pre- and post-monsoon than that measured using an eddy correlation approach over Lirung glacier (Langtang area, Nepal, 4250 m a.s.l., Steiner et al., 2018). Although the sites differ in topography, meteorology, and elevation and are, thus, not fully comparable, we suspect that sensible heat flux is overestimated by ISBA-DEB on West Changri Nup glacier.

Since $H = \rho_{air} * C_p * C_H * V * (T_{surf} - T_{air})$, where $C_p$ is heat capacity and $V$ wind speed, an anomalously large H implies that the surface debris in ISBA-DEB is overheating and evacuating too much heat. If any of the simulated incoming components to the energy balance are too large, the model could potentially compensate by overestimating the debris surface temperature. The overestimated temperature reflects the fact that multiple parameter sets can provide equally good model outputs (equi-finality principle) and could be due to a number of underlying factors. First, both the sensitivity tests and disagreement of ISBA-DEB's sensible heat magnitude and that of Steiner et al. (2018) suggest that $z_{o,m}$ =0.05 m and $z_{o,h}$ =0.005 m are not spatially representative values. Additionally or alternatively, processes missing from ISBA-DEB could influence H. ISBA-DEB's lack of advection (rather than explicit inclusion of wind dynamics like Evatt et al. (2015)) and its highly simplified vapor transport (rather than empirical fit to Monin and Obukhov (1954)'s curves) may account for the anomalously large H. Finally, we assume our observed energy fluxes comprise a closed budget, a condition that ISBA follows, but we cannot rule out that energy budget errors in observations contribute to the large H magnitude without a detailed future evaluation. Robust assessment of H over debris-covered glaciers requires more measurements using an eddy correlation approach.

Some decrease in sensible heat magnitude was achieved through adjusting the roughness lengths and albedo, although further work is necessary to improve the sensible heat flux calculated by ISBA-DEB when the simulated surface temperature

is greater than the prescribed air temperature for an extended period (i.e. during unstable conditions). Because an excessively large sensible heat flux removes heat from the debris that could otherwise be put towards glacier melt, resolving the sensible heat overproduction would likely lead to an increase in ISBA-DEB's glacier melt calculation.

## 5.4 Uncertainty

Although slope is known for West Changri Nup glacier's AWS, the slope at other sites where ISBA-DEB could be applied will inevitably vary. A slight change in surface slope, particularly if the slope is less than $5^o$, has a dramatic impact on the sub-debris melt calculated by ISBA-DEB. Runoff is directly proportional to slope angle such that a greater slope indicates more runoff and less potential for water buildup and turbulent heat exchange. A flatter slope gives a more water-saturated debris layer, and it's useful to make a comparison between the model runs with various slope values and model runs with dry vs saturated debris

(Section 5.2). For slopes greater than $5^o$, the debris is drained sufficiently well that it is no longer dominated by the thermal properties of water. Sub-debris melt on a slope of $0^o - 4^o$ somewhat resembles that under fully saturated debris, with the flatter debris having a lower interface flux and higher surface latent heat flux than the more sloped and comparatively drier debris. The flat debris does not show nearly the same magnitude of surface latent heat flux as the saturated debris does; while its top layer has more moisture than the debris on steeper slopes, it is far from fully saturated. The configuration that holds more water

in the debris has a greater in-debris latent heat flux (Figure 13f, like the lower panel of Figure 11).

    In order for melt computed by ISBA-DEB to be within 10% of true melt, albedo must not vary by more than $\pm 0.1$ and the thermal conductivity should stay within $\pm 0.15$ W m$^{-1}$K$^{-1}$. Table 4 shows the model sensitivity to roughness lengths and emphasizes the need to verify a site-specific value before applying ISBA-DEB at that different site. The varied measurements of momentum roughness length (2/3 on neighboring Khumbu glacier) increase model melt by 16, 30, and 39% over two years,

respectively, while the theoretically-reasoned greater roughness length decreases it by nearly 10% and Miles et al. (2017)'s value for boulders by over 30%. The thermal roughness length is even more poorly constrained than the roughness length for momentum, and our tests simply explored model response to a range of ratios. They demonstrate how crucial this parameter, which determines calculated latent heat fluxes, is to an energy balance model. Losing more energy to latent heat leaves less for glacier melt. Increasing the ratio of momentum to thermal roughness lengths from 10 to 25 increases melt by more than 10%.

Another source of uncertainty stems from the fact that the debris temperatures come from sensors which can migrate within the debris: the four sensors were installed at depths of 5, 7.5, 10, and 12.5 cm in 12.5 cm of debris in December 2012. Before the end of their deployment in November 2014, their depths were checked and reset only twice, December 2013 and April 2014. Therefore, while a portion of the modeled-measured temperature mismatch (Figure S3) is due to shortcomings of ISBA-DEB, another portion is due to the migration of the thermistors in the debris, which renders their depths unknown. It is not

possible to attribute the disagreement of ISBA-DEB temperatures with measured ones entirely to the model.

## 5.5 Future Directions

A central part of the ISBA structure is the neglect of advection based on the observation that advective heating makes relatively small changes to the soil temperature compared to conduction. In addition to the thermal properties, hydraulic properties, and

hydrological processes accounted for in ISBA-DEB, soil and debris also differ in the size of their interstitial void spaces. In highly permeable debris, there is ample space for air flow through the debris layer. Advective heat transfer is not accounted for in ISBA or ISBA-DEB.

Reznichenko et al. (2010) showed in a laboratory that rain advects heat from warm, highly permeable debris to the glacier surface. Sakai et al. (2004) showed that heat flux from percolated water assigned the temperature of debris was only 9% of the icemelt flux despite the fact that 75% of rainfall percolated, whereas the evaporative flux equaled nearly half of the net radiative flux, the main driver of glacier melt. They concluded that not accounting for the evaporative heat flux would lead to a twofold overestimation of sub-debris melt. They also pointed out, in comparing their two data collection sites, that, in contrast to soil, supraglacial debris has a higher permeability and lower evaporation rate. A lower evaporation rate is consistent with the fact that debris stores moisture at depth. Moisture deep in debris is less prone to evaporation than moisture on the surface since permeability (and, thus, ease of air flow) generally decreases into the debris (Evatt et al., 2015), although some does evaporate.

Evatt et al. (2015) designed a model that was novel in its incorporation of the evaporative heat flux throughout the debris layer. They justified their parameterization by noting that the melt occurs at the debris base and pointed out that calculating evaporation lower in the debris requires accounting for the air flow within the debris. Significant air flow is absent in soils for which the original ISBA was designed. By adjusting the surface vapor pressure term based upon the water content in the debris, Collier et al. (2014) also considered the evaporative heat flux within the debris layer.

Evaporative heat fluxes in ISBA are computed based on moisture content of only the top layer. While such a parameterization may be reasonable in soils, it has no physical basis in far more permeable debris, within which the atmosphere can exchange heat and mass at different depths. Collier et al. (2014) found that the latent heat flux calculated at the saturated horizon of their reservoir model was too low, offering that their computed saturated horizon itself was flawed without accounting for capillary effects. Therefore, ISBA-DEB provides an advancement in prognosing the location of moisture, which both pools (as in Collier et al. (2014)'s model) and undergoes diffusion. Introducing the possibility for latent heat fluxes to arise from within the debris is the next step in advancing ISBA-DEB.

Simulating neither evaporation nor advective heat transfer within the debris accounts for the relative melt modeled under 12.5 cm of wet, dry, and partially wet debris (Section 5.2, Table 3). While ISBA-DEB can accommodate any debris thickness, so long as the entire debris-glacier column is discretized into 20 or fewer layers, we expect sub-debris melt values calculated by the model in its current version to match measurements of melt under thinner debris more closely than measurements under thicker debris. The effect of neglecting evaporation deeper than the surface is decreased for thinner debris, as is the effect of neglecting advection within the debris. With thicker debris – or, more generally, with debris covers in which sub-surface evaporation and/or advection are more favorable – we expect the departures from measured sub-debris melt values due to these unrealistic assumptions in ISBA-DEB to be more pronounced.

Correct representation of snow is extremely important to ISBA-DEB's performance. Snow is a highly-reflective, strong insulator, and any error in simulated occurrence of snowfall will cause error in not only the surface energy balance and underlying debris temperature profile simulated by ISBA-DEB (e.g. Figures 5, S3) but also the water mass budget of the debris. An error in snow cover timing or duration affects the net radiation budget and could potentially contribute to the model's overestimation

of sensible heat flux (Section 5.3). In this study, we used precipitation data from Pyramid Research Station and partitioned phase based upon AWS air temperature. The SR50 depth sensor provides additional information: it may indicate snowfall when there was no recorded precipitation at Pyramid, and it may conversely indicate no solid precipitation when subfreezing temperatures at the Changri Nup AWS coincided with recorded precipitation at Pyramid. Any remaining mismatch after the basic site-specific adjustments performed in this study would propagate error to the calculated ablation. Verifying snow cover duration in the forcing is, therefore, an important undertaking in future research using ISBA-DEB on Changri Nup glacier and should be a priority when collecting data with which to force a debris-covered glacier surface energy and mass balance model – particularly if it includes moisture.

Finally, with spatially distributed, glacier-wide modeling in mind, our sensitivity tests show that it is important to investigate not only values of roughness lengths that govern the turbulent heat fluxes but also the spatial distribution of those values. Albedo and emissivity are of lesser importance to constrain beyond their plausible values. The model's sensitivity to thermal conductivity and slope perturbations, particularly on flatter terrain, reflects its sensitivity to water content. It is, therefore, important to compute thermal conductivity correctly, taking thermal conductivity measurements for dry debris as well as investigating how varying amounts of liquid water change the bulk value. Roughness lengths, dry debris thermal conductivity, and slope are enormously important variables to constrain when performing distributed surface energy balance modeling over a debris-covered glacier (the slope controls the water content enough such that, with an accurate measure of dry thermal conductivity and a known relationship between moisture and bulk conductivity, spatially distributed values of thermal conductivity can be calculated from slope). Debris surfaces are typically very rough, with variable slopes over short distances. As a result of this topography, it is common to see saturated debris and pooled water in topographic lows and dry debris on topographic highs. Slope is also crucial for identifying low-lying troughs, where pooled water or saturated debris could dominate the surface albedo and emissivity (e.g. lakes over supraglacial debris have lower albedo, Miles et al., 2016). Distributing ISBA-DEB would, then, produce large spatial variability of melt and sublimation at the glacier scale because of the large variability of not only debris thicknesses and properties (Nicholson et al., 2018; Rowan et al., 2017; Rounce et al., 2018) but also of the water content in debris. Accounting for advection may change the model simulation of glacier ice melting most under completely dry debris.

## 6    Conclusions

While the introduction of advective heat transfer and atmospheric exchanges deeper than the surface of the debris could make the model more physically realistic, ISBA-DEB nevertheless provides an advancement in modeling the processes in a debris layer. It reasonably simulates the temperature evolution of a snow-debris-glacier column according to meteorological forcing and evolving thermal properties year-round, even when the ice temperature is subfreezing and a snowpack is present on the debris. It successfully produces variations in non-saturated water content, phase, and location, demonstrating both diffusion and water pooling at the glacier surface. It also computes glacier melt based on the processes of heat and water transfer, their determination of thermal and hydraulic properties, and their interplay with one another.

ISBA-DEB is the first debris surface energy balance model to integrate heat conduction with moisture diffusion. In its simulations of West Changri Nup glacier, enhanced melt occurs below dry debris due to a combination of greater thermal diffusivity and little loss of energy to evaporation or sublimation. ISBA-DEB explicitly accounts for the atmosphere-debris latent heat exchanges in the top (surface) layer of the debris only. The large difference in glacier melt below dry and saturated

debris shows that latent heat is enormously important in removing energy from the system. Accounting for moisture in the conductive heat flux alone is insufficient when modeling melt under a debris-covered glacier. It is, therefore, an essential next step to examine and incorporate the latent heat exchanges of moisture at all depths in the debris.

ISBA-DEB provides a basis for developing a model that can be applied at the glacier scale by identifying not only the importance of atmospheric exchanges throughout the debris column but also the most sensitive parameters controlling the melt

at point scale. In addition to using accurate roughness lengths, it is crucial to represent moisture sources and sinks correctly. An important part of the latter is constraining the lateral runoff timescale (through, for example, laboratory or field-based experiments).

ISBA-DEB may be used to explore past or future changes in sub-debris melt. Reanalysis data, such as that of ERA Interim, provides all variables necessary to drive the model. Running ISBA-DEB under various Representative Concentration Pathway

(RCP) emissions scenarios (Van Vuuren et al., 2011; Allen et al., 2014) would provide insight into the fate of ice under debris, an increasingly important topic as debris cover is increasing in a warming climate (Thakuri et al., 2014; Kirkbride and Deline, 2013).

*Code availability.* <surfex_git2 @ V8_1_giese> code available at:

opensource.umr-cnrm.fr/projects/surfex_git2/repository?utf8=?&rev=V8_1_giese

*Data availability.* FORCING.nc and OPTIONS.nam files available at https://glacioclim.osug.fr/spip.php?article75&lang=fr

| Model Parameter or Physical Constant | Value | Source/Note | Where specified |
|---|---|---|---|
| Ice density (kg m$^{-3}$) | 917 | | |
| Air density (kg m$^{-3}$) | 0.644 – 0.720 | | |
| Air thermal cond. (W m$^{-1}$ K$^{-1}$) | 0.024 | introduced in ISBA-DEB, Haynes (2017) | |
| Air heat capacity (J kg$^{-1}$ K$^{-1}$) | 1006 | introduced in ISBA-DEB, Haynes (2017) | |
| Water density (kg m$^{-3}$) | 1000 | | |
| Water thermal conductivity (W m$^{-1}$ K$^{-1}$) | 0.57 | | |
| Water heat capacity (J kg$^{-1}$ K$^{-1}$) | 4218 | | |
| Ice thermal conductivity (W m$^{-1}$ K$^{-1}$) | 2.22 | | |
| Ice heat capacity (J kg$^{-1}$ K$^{-1}$) | 2106 | | |
| Gravitational acceleration (m s$^{-2}$) | 9.80665 | | |
| Latent heat of vaporization of water ($L_v$) (J kg$^{-1}$) | $2.5008 \times 10^{6}$ | | |
| Latent heat of sublimation of water ($L_s$) (J kg$^{-1}$) | $2.8345 \times 10^{6}$ | | |
| Latent heat of fusion of water ($L_m$) (J kg$^{-1}$) | $3.337 \times 10^{5}$ | | |
| Debris emissivity ($\epsilon$) | 0.94* | Reid and Brock (2010) | OPTIONS & modd_isba_par.F90 |
| Debris albedo ($\alpha$) | 0.2* | calc. from SW measurements | OPTIONS |
| Dry debris density (kg m$^{-3}$) | 1690 | measured | modd_isba_par.F90 |
| Dry debris thermal cond. (W m$^{-1}$ K$^{-1}$) | 0.94* | Reid and Brock (2010) | thrmcondz.F90 |
| Dry debris heat capacity (J kg$^{-1}$ K$^{-1}$) | 948 | Reid and Brock (2010) | modd_isba_par.F90 |
| Debris vol. heat cap. (J m$^{-3}$ K$^{-1}$) | 1602120 | - | - |
| Debris saturated hydraulic conductivity ($k_{sat}$, m s$^{-1}$), Layers 1 – 12 | 0.03 | Domenico et al. (1998) | init_veg_pgdn.F90 |
| Debris saturated hydraulic conductivity ($k_{sat}$, m s$^{-1}$), Layer 13 | 0 | | init_veg_pgdn.F90 |
| Matric potential at saturation ($\Psi_{sat}$, m) | 0.097** | | init_veg_pgdn.F90 |
| b | 3.8** | | init_veg_pgdn.F90 |
| Debris porosity ($\Phi$) | 0.388465**, 0.999 | $\sim$ 0.37 (measured) | init_veg_pgdn.F90 |
| Debris surface $z_{om}$ (m) | 0.05* | Lejeune et al. (2013) | OPTIONS |
| Debris surface $z_{om}/z_{oh}$ ratio (m) | 10* | ISBA default, Mascart et al. (1995) | OPTIONS & ini_data_param.F90 |
| Shape factor | 30 | tuned | hydro_soildif.F90 |
| $\tau_{max}$ (s) | 86400 | tuned | hydro_soildif.F90 |
| $\tau_{min}$ (s) | 3600 | Collier et al. (2014) | hydro_soildif.F90 |
| Slope ($^{o}$) | 5* | measured | hydro_soildif.F90 |
| $\alpha_\wp$ | 6 | Johnsson and Lundin (1991) | hydro_soildif.F90 |
| $w_{min}$ (m$^{3}$ m$^{-3}$) | 0.0001 | 0.001 in ISBA | modd_isba_par.F90 |
| Model time step (s) | 900 | with splitting in hydro_soildif.F90 | OPTIONS |
| Number of calculation layers | 13 debris, 7 ice | measured | OPTIONS |
| Debris layer thickness (cm) | 12.5 | measured | |
| Altitude of measurement site (m) | 5360 | measured | FORCING |

**Table A1.** Physical constants as well as parameter values used in the baseline ISBA-DEB; sensitivity tests were performed on parameters with a single asterisk. Double asterisks appear with values predicted by pedotransfer functions (PTFs) of Noilhan et al. (1995) (using Clapp and Hornberger, 1978) based upon an input of 98% sand and 2% clay. The calculated porosity given by the PTFs is 0.39, close enough to the measured porosity of 0.37 that we did not overwrite the PTF calculation. The designation of zero hydraulic conductivity of the bottom debris layer simulates an impenetrable glacier surface and ensures no non-physical drainage out of the debris into the glacier. The third column of the table indicates the file in which these parameters are set, for the future user. Air density is a function, as described in the caption of Table S1. Values for which no references are listed are the standard values used by SURFEX (LeMoigne, 2018).

*Author contributions.* A. Giese conceived the study, performed the modeling work, and wrote the manuscript. A. Boone assisted in modeling work; he was instrumental in the coding of ISBA-DEB adaptations and in integrating them into ISBA. P. Wagnon made the meteorological forcing data and debris temperature data available for this study, and he provided invaluable feedback on the direction of the modeling work and on the specific model output. R. Hawley helped troubleshoot many modeling hurdles and advised on the methodology for the tuning and sensitivity tests.

*Competing interests.* There are no competing interests.

*Acknowledgements.* S. Morin provided an introduction and orientation to using SURFEX, and M. Lafaysse (Météo-France - Centre d'Etudes de la Neige, CEN) and S. Schwarz (Dartmouth) answered many technical questions. Y. Lejeune corrected precipitation data and together with M. Dumont facilitated A. Giese's work during a 10-month stay at CEN. G. Lewis provided valuable feedback and, along with J. Chipman, created the map. C. Vincent offered advising throughout the project, and Y. Arnaud helped define a direction and also gave feedback. We thank S. F. Sherpa, D. Shrestha, C. Vincent, Y. Arnaud, F. Brun, J. Shea and many other people who cannot be all listed for their assistance in the field. This work has been supported by the French Observatory CRYOBS-CLIM now part of the IR OZCAR, the French National Research Agency (ANR) through ANR-13-SENV-0005-04-PRESHINE, and by a grant from Labex OSUG@2020 (Investissements d'avenir - ANR10 LABX56). This study was carried out within the framework of the Ev-K2-CNR Project in collaboration with the Nepal Academy of Science and Technology as foreseen by the Memorandum of Understanding between Nepal and Italy. Thanks to contributions from the Italian National Research Council, the Italian Ministry of Education, University and Research and the Italian Ministry of Foreign Affairs. We thank Pyramid staff and Glacier Safari Treks for their logistical support. A. Giese was supported by NASA Space Grant NNX15AH79H and NSF GRF grant DGE-1313911.

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
