# Peer review of "Incorporating moisture content in surface energy balance modeling of a debris-covered glacier"

_The Cryosphere, 2019_

## Referee Comment (RC1) · Anonymous Referee #1 · 21 Nov 2019

Review of Giese et al. (2019) Incorporating moisture content in surface energy balance modelling of a debris-covered glacier

Giese et al. present the adaptation of the ISBA model to represent supraglacial debris, and hence allow the calculation of sub-debris melt using this model. The key advance is the incorporation of moisture within the debris, which can vary over time and with depth, and which then influences the thermal properties of the layer and hence the latent heat flux and melt. The authors test their model on West Changri Nup Glacier and conduct an uncertainty analysis by varying parameters based on ranges found from the literature. They also conduct runs under varying amounts of debris saturation. They describe the performance of the model and discuss the processes it includes compared to other models of sub-debris melt.

General comments:

It is certainly a good advance to include the moisture explicitly in a model of sub-debris melt and modifying ISBA to do this seems like a good idea and something I would like to see published. The paper is well written on the whole and explains the model advance quite well. Its concisely written and the overall approach seems robust. There are likely still some parts of the model which are less like the reality in the debris (e.g. the overestimation of the sensible heat flux and possibly moisture being held in the lowest debris layer) but nevertheless this is an important step forward.

In places the paper can lose clarity a little, and could do with explaining some parts of the model, and especially the collected field data more explicitly (see specific comments for details). Moreover, I think a key thing which is missing is the ability of the authors to really bring out the understanding of the interplay between the energy and moisture fluxes and the changes to the debris properties and how this affects melt overall. This could be strengthened in the results and discussion by better explaining the links between the debris properties and moisture content when explaining the results (improving some of the figures would help in this regard too). For instance, a clear explanation is missing of how the moisture content, thermal conductivity and heat capacity evolve on a daily and seasonal scale, of why moisture tends to be concentrated in the lowest layer and what the explicit differences are between the dry and saturated conditions.

Also, although the future directions section compares the results with the broader literature, this could be strengthened by having a clearer uncertainty analysis (and hence clearer recommendations for where data and understanding is lacking) and a clearer idea about what steps would be needed to bring this modelling from the point to the glacier scale. Furthermore, do you think that the findings determined for this glacier would hold for others, for instance with finer or thicker debris? At the moment the paper presents the new modelling (which is great to see), but I think it would really increase the usefulness of the paper if it could go beyond this a bit more to both better explain

the findings and think more broadly about the consequences of those findings. I would also say that the key 'take home messages' of the paper are not so clear. I feel less sure about what the new insights are which the modelling has made possible, the authors could therefore make this much clearer in the results/discussion and conclusions.

One other point is the use of the Reid and Brock (2010) parameters for dry debris, when the values were measured under conditions when there was likely melting at the ice-debris interface, and so the values would be for partially saturated conditions.

Specific comments:

Pg 2 Line 19: Reid and Brock don't necessarily assume 'dry' debris, just assume the debris characteristics are constant and the same as the average measured conditions.

Pg 2 It would be worth mentioning Evatt et al. (2015) Glacial melt under a porous debris layer, somewhere in the introduction (likely in the paragraph beginning on line 18), given their inclusion of the evaporative heat flux at the bottom of the debris layer.

Table 1: It would be better if the equations and the explanation in the caption could be taken out of this table and moved to Supplementary material. It is not clear where the values for ice and water conductivity and volumetric heat capacity are from (even if they are standard values). Given the importance of these parameters it would help to clearly explain the relation between them in the introduction. Also, it is unclear why the Reid and Brock (2010) value of thermal conductivity is used for dry debris, given that in the caption it is termed an 'effective' value from the time of measurements (this is confirmed in Brock et al, 2010). It is likely that this value is for a partially water saturated debris layer, rather than a dry one. Even if you decide this is a value for 'dry debris' then I think you would need to know the porosity of the debris on Miage Glacier to determine the 'debris' thermal conductivity so that you could then calculate water and ice saturated debris. I am not convinced therefore that the calculation of the water and ice saturated debris is correct. Fully name the parameters, e.g. Thermal conductivity, Specific heat capacity.

Filed site: it would be good to get an idea of the debris thickness, grain size and geology for West Changri Nup.

Figure 1: This map could be improved. The text on the inset map is unreadable, and the inset needs a scale bar. The main pane is too zoomed in, it would be better to show the full outline of both glaciers and label them.

Pg 5 Line 23: 'air temperature, humidity and surface fluxes. . .' Also, for clarity would be better to state the surface fluxes measured (incoming and outgoing short and longwave radiation).

Figure 2: Is the heat flux due to precipitation calculated?

Pg 6 line 1: Is the surface debris temperature calculated, or taken from measurements?

Pg 6 line 8 It is not clear what data is used for the spin up, 40 x the met data?

Pg 6 How many layers are for snow?

Figure 3: Give the symbols for the input variables within this figure (especially the top right bubble), so it is absolutely clear what the input variables are. In the third box down, surely all the fluxes (not just latent heat) are calculated?

Pg 8 line 6: Refer here to the supplementary material where you give the full energy balance.

Equation 3: Should the wi not be wmin in the brackets?

Pg 8 Line 27: 'where Lm is the. . .' Also it would be useful to explain in words what each part of these equations represent. I'm a little unclear about the source and sink terms and how they represent the system.

Pg 9 Line 10: The lack of ice growth in subfreezing temperatures may suggest a lack of vapour transport, but what about in above freezing temperatures?

Pg 9 Line 14: How was the isothermal vapour conductivity derived, or what value was

used?

Pg 9 Line 15: The observations of moisture transport, are these yours, or another author? It would be interesting to know of measurements of moisture transport if these are available. If based on your observations then that's fine but make this clear.

Pg 9 Line 24 Should there not be brackets around wi/w (following Boone et al., 2000)?

Pg 9 Line 32: It would be worth mentioning that debris tends to coarsen upwards (e.g. as mentioned by Reid and Brock, 2010).

Pg 10 Line 6-7: Do you mean with depth of water or with depth in the debris? And why would it increase with depth (of debris, especially?)

Equation 8: I can't see $\Delta z_j$ or $w_j$ defined anywhere? (I think the subscript j just needs defined somewhere, unless I've missed it)

Figure 4: Is this depth of the debris or water depth on the y-axis?

Section 4: It would probably be more useful to have the forcing before the model, although I understand that you are presenting a new model method. It is just easier to follow. The meteorological variables could easily be listed or put in a table (maybe refer to Table 2), there can't be that many. The section on in-situ meteorological measurements is lacking detail. How was the surface debris temperature situated (debris surface temperatures are easily overestimated), also give the depths of the sensors? How was ablation measured (with a UDG?) Give a reference for the ISBA-DEB being insensitive to the $CO_2$ flux. The point about the precipitation should be moved to section 4.2 (in fact the section headings 4.1, 4.2 and 4.3 could easily be removed). How was debris density and porosity measured?

Pg 12 Line19: When does the precipitation dataset end? How was the precipitation at Pyramid corrected (or give a reference that explains)?

Pg 15 Line 21 Give the range in error of $\tau\alpha$ and $\tau$max to show that the actual error

values were similar.

Figure 6 it would be more useful to show the mean diurnal values, or at least only the time when both are compared within the same depth, with measured and modelled on the same panel. This would make it much easier to compare the measured and modelled outputs.

Figure 7 Text on these figures could be a little larger. On b) consider including the air temperature and debris surface temperature.

Pg 17 Line 8 The above freezing temperatures propagating into the ice is shown better in Figure S2.

Figure 8 Really Figure S2 is more useful than this figure, consider swapping the two.

Page 18 The description of the change in the debris moisture content is a bit too terse. It would be helpful to describe this more thoroughly, especially as this is a key new function of the model. How does the thermal conductivity and heat capacity vary with the moisture content on a daily and seasonal scale? Also, explain why only the bottom layer is holding water.

Figure 9 It would help to add side panels to both a) and b) of the diurnal average values for each layer (perhaps just during melting). This would help interpretation.

Figure 10 This would be fine as a table, with the 2014 results given too. I am surprised that the results for the full model run with the integrated moisture content is not included too?

Pg 20 Line 5 Compare the dry and saturated debris directly for clarity, i.e. the thermal diffusivity of dry debris is around double that of saturated debris, hence the higher melt under dry debris.

Pg 20 I don't think the model includes the wind dynamics within the debris (unlike Evatt et al., 2015), would doing so change these results?

Figure 11 Again it would be helpful to show the mean diurnal patterns in each layer as a side panel, especially within the ablation season.

Figure 12 Consider changing the upper panel title to 'Debris surface latent heat flux from atmospheric exchange'. Also label panels a) and b) as for the others.

Page 22 Line 5 Including debris moisture does not significantly decrease sub-debris melt – compared to what? (the dry scenario?)

Pg 22 Line 8 Can you explain why only the lowermost layer holds water, is this to do with the runoff parameterisation or the debris porosity?

Pg 22 Line 19 Figure 12 is missing the cumulative flux used for melt.

Pg 23 Line 3 In Figure 10 though it seems to show similar melt between dry and partially debris?

Pg 23 Lines 15-27 Consider just citing the papers in Table 3 to shorten this section.

Table 3 Just give the symbols rather than the shortened variable names in the first column. It would also be helpful to know the % change of the parameters, otherwise its tricky to determine which parameters the model is most sensitive too. I understand the overall rationale of using the literature values to give sensible ranges but this would help clarify the sensitivity of the model.

Pg 24 Line 1 For the extreme values tested, is this with the other parameters at their * value?

Pg 25 Lines 1-3 A more thorough description of which variables the model is most sensitive to is needed here (as mentioned would help to show the % change in each variable) and an idea of which ones are most likely to be well or not well known. Then suggest where work should concentrate to improve the knowledge of the most sensitive variables, including of their spatial distribution.

Pg 27 Lines 6-16 This very high surface temperature could be indicative of the debris

properties not being correct, as the model might be compensating by increasing the debris surface temperature. I am presuming here that the debris surface temperature is modelled (I think it is, although I am not sure if this explicitly stated).

Pg 28 Lines 1-2 Here it would be useful if the authors could comment on what they think would happen over a real debris-covered surface where there is a combination of steep and shallow slopes. What would likely be the overall affect? On line 1 specifically the authors mention' overlying flatter glaciers' which is a strange term, say specifically flatter slopes, as the slope angle will vary across the glacier surface (especially for debris-covered glaciers).

Pg 28 Line 11 what ratio are you referring to here?

Pg 28 Line 33 Explain why moisture deep in the debris is less prone to evaporation. (Possibly because of lower wind speeds and cooler temperatures at depth, but explain this based on your results).

Pg 29 Line 23 'snow melt rate' ? Also consider 'measured snow melt' instead of 'the SR50 data' just so it is clear what you are meaning without readers knowing your sensor names.

Table A1 I still have concerns over using the Reid and Brock (2010) values of debris thermal conductivity and heat capacity for dry debris. There are no values given for the matric potential at saturation or b (what is this?). I understand they are predicted values, but does this mean they change over time if one value isn't given?

Supplementary Material

I might have missed it but please refer to the energy budget section in the main paper. Also, it would be helpful to include the equations for the sensible and latent heat fluxes, since these can vary between models.

Pg 3 Equation 13 LEgu I think should be LEgi (or vice versa) to match the text.

Pg 3 Equation 15 LEgf I think should also be LEgi? I can't see Wn defined, should this be the change in water stored in snow (Ws)?

Pg 3 Line 15 is Ps the melted snow in water equivalent?

Figure S2 This is a useful figure and I would suggest including it in the main text. It would be useful to have the rain and snowmelt also in mm (to match glacier melt). Rename Layer 13 as the ice surface (rather than glacier surface). Write 'Temperature' rather than just T in the legend, and say ice content and water content, rather than just ice and water, for clarity. Ideally the middle four panels should have the same y-axis scale for comparison. Alternatively show all 4 layers of water content together in a pane, with separate panes for ice content and temperature.

Technical corrections:

Pg 1 line 12 'debris-covered'

Pg 4 Line 3 and line 9 'debris-covered'

Pg 4 line 9: 'mostly composed of clean ice.'

Pg 12 Line8: delete 'dot in'. 'AWS data was measured half hourly from the 6th December 212 15:00 – 28 November 2014 13:30 local Nepal time, and provided'

Pg 15 Line 18 'measured ones from'

Pg 17 Line 2 'Figure S2'

Pg 25 last paragraph, join this to the next paragraph.

Pg 27 Line 24 'using an eddy correlation approach'.

Pg 28 Line 4 'thermal conductivity'

Pg 28 Line 5 'varied measurements of roughness length' (I am presuming here)

Pg 28 Line 12 'Measured ablation over the two years modeled' Strange phrasing here,

reword.

Pg 28 Lines 12-19 Consider shortening this section.

Pg 28 Line 30 'in comparing their two data collection sites,'

Pg 29 Line 20 'It also computes glacier melt'

Figure S1 Add a y-axis title 'Energy flux (W m-2)'

---

## Referee Comment (RC2) · Anonymous Referee #2 · 28 Dec 2019

Comment on 'Incorporating moisture content in surface energy balance modeling
of a debris-covered glacier ' by Giese et al.

In this paper, authors applied ISBA model to debris-covered ice. The model include moisture transport and phase changes in the debris layer. They used observed temperature in the debris layer and melt amount in 2013 as validation. Then, they also carried out sensitivity test for six elements.

But, I think some explanation and discussion are not sufficient in the manuscript. I hope my comment will help to improve your paper.

Main comments

1. There is no 'observed data' section. Observed glaciological or meteorological data are important to establish model as not only input data (AWS) but also validation (ablation). Even some data are already published, I think authors need to describe what kind of observed data authors have. I recommend to add one section of observation.

2. Authors describe in the conclusion 'Snow is a strong insulator, and any error in simulated occurrence of snowfall will cause error in the surface temperatures and underlying debris temperature profile simulated by ISBA-DEB (e.g. Figure 6).' In P29 Line24. But, they did not discussed about the snow cover effect in the discussion. Snow cover makes high albedo over the debris, which inhibits ice melting. Snow cover makes no temperature gradient in the debris layer. And meltwater from the snow cover will be important moisture source of debris layer. Therefore, reconstruction of snow cover duration is significant to estimate ablation under the debris. Authors have nice observed data, and simulated data, then they can discuss about the snow cover effect.

Specific comments

P3 Table 1 > multiplication should be presented as '×' not '*'.

P4 L9 > I think 'It lies 200 m southeast of' should be southwest. or I misunderstand them?

P4 Fig.1 > There are no description of the location of "West Changri Nup and North Changri Nup glaciers' in the map. So, I cannot detect those locations. Further, location of measurement site of debris temperature are necessary.

P5 L22 'Glacier melt enters the debris at the base, and rain and snowmelt enter the debris at

the surface.' > better to add water like 'Glacier melt water, snowmelt water'

P6 Figure 2 > The small letters (Ex. dz varies, n layers) in the left side of this figure can not be read, when the figure has been shrunk.

P7 Figure 3 > I found several 'Table A1' in the manuscript. But, there is no table that shown as 'Table A1'. Revise all 'Table A1'.

P8 L7-8 'We neglect energy carried by precipitation' > But, there are arrow of 'Rain &snowmelt' in the Figure2. Please explain the detail, how did you treat the penetrated water

P10 L11 '$\tau_\alpha$... defining the runoff timescale from....' > $\tau_\alpha$ is important in this paper. Then, I recommend that detail explanation of $\tau_\alpha$ are necessary here. For example, *** become larger with increasing $\tau_\alpha$ and so on.

P11 Figure 4 I recommend to change the unit of $\tau_\alpha$ in hour rather than in second, because in the text you have discussed the $\tau_\alpha$ in hour.

P12 Line5 I recommend that all meteorological variables are described in the supplement.

P12 L13 There are no description about how did you measure the debris temperature and debris density and porosity.

P12 L21 I estimate you used eq.(1) in Wagnon et al.(2009) here. But please draw the equation between -1 to +3 ºC. The equation has no simple relation. I can't imagine why you have to use the complicate relation. Or you have applied for the temperature range between 0 and +2 C ?

[Figure]

P12 L22 'minor adjustments based on SR50 measurements.' > There is no detail information on the adjustments. I can estimate that measurement of precipitation using Geonor T200B has relatively small error because it can measure weight of precipitation directly. On the other hand SR50 measure only surface elevation change, which is not equal to amount of precipitation. So, why you should adjust using SR50 ?

P13 Table 2 You should write name of instrument at the 'Instrument' for example Pyranometer for shortwave radiation.
Campbell SR50 can not measure ablation or accumulation directly. It can measure only elevation change, then, you should assume snow or ice density to estimate accumulation or ablation.

P14 Figure 5c It seems that many point of rainfall have been overlaid by those of snowfall. I think these two data should be shown separately. The label of x-axis should rotate 90 degree. Vertical writing for label of x-axis is easy to detect the exact location.
There are rainfall, but no snowfall during winter (from Jan 2014 to around March 2014 ). Is that possible? Please check.

P15 L17 '2013 ablation' > There is no information of the observation of 2013 ablation. Location? Duration? Frequency of measurement ? How much is the debris

thickness? How did you measure? And the location of ablation measurement were same with that of measurement of debris temperature ?

P15 L18 'using an RMSE calculation to capture the magnitude of temperature.' >> I recommend to write them specifically.

P16 Figure 6 I recommend that daily average data both observation and calculation are plotted in one graph. I cannot compare if those data are shown in different graph. Further, many points are overlapped in Fig. 6, then, daily average are necessary. And I also recommend that scatter plot of the observed and simulated debris temperature are valid to compare and analyze.
And how much is the debris thickness? 12.5 cm ?

P16 Figure 6 '(having insufficient snowfall to produce the observed persistent snowcover)' > It might be possible that model overestimate the melt rate of snow cover. I think reconstruct of duration of snow cover during the melting season is significant (see main comment).

P17 Figure 7 In the explanation, '(a) Temperatures in debris (top 12.5 cm)' I can't understand the 'top 12.5 cm'. This points location of interface between debris and ice ?

'… below the black line indicates…' > 'below' ?

P17 L3 Figure A2 > Figure 2

P 18 Figure 8 I recommend to describe those data is simulated data.

P20 Figure 10  There is no calculated data shown in Figure 8 (calculation of moisture integrated model). Why? It seems that the calculated ablation using moisture integrated model (Figure 8) have similar value with 'dry' and 'partially saturated'. And in this section (P20 L1-8), there is no discussion about the difference between three case calculated data and measured data. Measurement value was corresponded with the value of assuming fully saturated, not dry or partially saturated.

P21 Figure 11 This is a minute thing、but as for color bar, 'No moisture' are colored by light red. I think the color should be white.

P22 Figure 12 I recommend to add cumulative melt (like Fig. 8).
In the last sentence of the explanation, there are 'A greater latent heat flux corresponds to a lower conductive heat flux through…' > There are multiple latent heat fluxes in this system. Please write specifically.

P24 Table 3 '% Change' > What is the change ? Change of ablation ?

P28 L12 '75.3±20 cm (2012 – 2013) and 47.1±20 cm (2013 – 2014)' > Here, I have noticed that you have observed data not only 2012 – 2013 but also 2013 – 2014. Why you did not compare observed data and simulated data 2013 – 2014? Because you have tuned parameter 2013 – 2014? But, you have tuned parameter $\tau_\alpha$ fitting simulated debris temperature to observed debris temperatures. Not ablation amount.
Observed data of ice melt under debris layer are very few, even though those data have large uncertainty.

P29 L23 'more detailed assimilation of the snow rate with the SR50 data.' > Still I don't know the location of the measurement of ablation and debris temperature. Then, my comments might not appropriate. I would like to ask the surface temperature at AWS (SR50 measurement) can be assumed to be same condition with that at ablation measurement site? If you can assume same condition, you can analysis the albedo of AWS using observed SRs downward and upward and weather there is snow cover or not.

P29 L24 'Snow is a strong insulator, and any error in simulated occurrence of snowfall will cause error in the surface temperatures and underlying debris temperature profile simulated by ISBA-DEB (e.g. Figure 6).' > I recommend to discuss about the snow cover during summer season in the discussion section. Because you have observed data of temperature of debris as shown in Fig.6.

P29 L26 'ISBA-DEB may be used to explore past or future changes in sub-debris melt. Reanalysis data, such as that of ERA Interim, provides all variables necessary to drive the model.' > ERA Interim do not include debris-thickness or particle size of

debris, which is important for permeability (thermal conductivity) of debris and also surface roughness, those parameter have high sensitivity in ISBA-DEB model in your result. How will you apply the ISBA-DEB model to other debris-covered area? I recommend that you have to better to consider application to other debris covered part.

---

## Author Comment (AC1) · 25 Jan 2020

Comment on 'Incorporating moisture content in surface energy balance modeling of a debris-covered glacier' by Giese et al.

In this paper, authors applied ISBA model to debris-covered ice. The model include moisture transport and phase changes in the debris layer. They used observed temperature in the debris layer and melt amount in 2013 as validation. Then, they also carried out sensitivity test for six elements.

But, I think some explanation and discussion are not sufficient in the manuscript. I hope my comment will help to improve your paper.

*Thank you, these comments have definitively helped to improve the paper.*

[Figure]

Main comments

1. There is no 'observed data' section. Observed glaciological or meteorological data are important to establish model as not only input data (AWS) but also validation (ablation). Even some data are already published, I think authors need to describe what kind of observed data authors have. I recommend to add one section of observation.
*The paper structure has been rearranged to include "In Situ Measurements" and "Model Inputs" subsections within Section 2: "Field Site and Data," before the model description (Section 3), including all information from former sections 4.1-4.3.*

2. Authors describe in the conclusion 'Snow is a strong insulator, and any error in simulated occurrence of snowfall will cause error in the surface temperatures and underlying debris temperature profile simulated by ISBA-DEB (e.g. Figure 6).' In P29 Line24. But, they did not discussed about the snow cover effect in the discussion. Snow cover makes high albedo over the debris, which inhibits ice melting. Snow cover makes no temperature gradient in the debris layer. And meltwater from the snow cover will be important moisture source of debris layer. Therefore, reconstruction of snow cover duration is significant to estimate ablation under the debris. Authors have nice observed data, and simulated data, then they can discuss about the snow cover effect.
*This is a valid point, and investigation into the effect of any error in snow cover is an important next step. The complexity of such an undertaking makes it beyond the scope of the paper; however, we have added the following to our Discussion in "Future Directions": "snow is a highly-reflective, strong insulator, and any error in simulated occurrence of snowfall will cause error in not only the surface energy balance and underlying debris temperature profile simulated by ISBA-DEB (e.g. Figures 6, S3) but also in the water mass budget of the debris. An error in snow cover timing or duration affects the net radiation budget and could potentially contribute to the model's overestimation of sensible heat flux (Section 5.3). In this study, we used precipitation data from Pyramid Research Station and partitioned phase based upon AWS air temperature. The SR50 depth sensor provides additional information: it may indicate*

*snowfall when there was no recorded precipitation at Pyramid, and it may conversely indicate no solid precipitation when subfreezing temperatures at the Changri Nup AWS coincided with recorded precipitation at Pyramid. Any remaining mismatch after the basic site-specific adjustments performed in this study would propagate error to the calculated ablation. Verifying snow cover duration in the forcing is, therefore, an important undertaking in future research using ISBA-DEB on Changri Nup glacier and should be a priority when collecting data with which to force a debris-covered glacier surface energy and mass balance model–particularly if it includes moisture."*

Specific comments
P3 Table 1 > multiplication should be presented as '×' not '*'.
*FIXED.*

P4 L9 > I think 'It lies 200 m southeast of' should be southwest. or I misunderstand them?
*FIXED.*

P4 Fig.1 > There are no description of the location of "West Changri Nup and North Changri Nup glaciers' in the map. So, I cannot detect those locations. Further, location of measurement site of debris temperature are necessary.
*FIXED.*

P5 L22 'Glacier melt enters the debris at the base, and rain and snowmelt enter the debris at the surface.' > better to add water like 'Glacier melt water, snowmelt water'
*FIXED.*

P6 Figure 2 > The small letters (Ex. dz varies, n layers) in the left side of this figure can not be read, when the figure has been shrunk.
*FIXED.*

P7 Figure 3 > I found several 'Table A1' in the manuscript. But, there is no table that shown as 'Table A1'. Revise all 'Table A1'.
*We confirm the presence of Appendix (not Supplement) Table 1. It was on page 31 of the original manuscript.*

P8 L7-8 'We neglect energy carried by precipitation' > But, there are arrow of 'Rain &snowmelt' in the Figure2. Please explain the detail, how did you treat the penetrated water
*We added an explanation for neglecting energy carried by precipitation and added clarity to the figure, which has arrows for the precipitated mass entering the system.*
*In SURFEX, there is an option to include the energy of rainfall if it is above 0C. Generally, this option is not activated since it cannot currently be used in the coupled (land-atmosphere) configuration because it would require removing this energy from the atmosphere. In other words, it would require keeping track of the temperature of falling precipitation rather than using the current assumption used in most operational and climate models that the rainfall has the same temperature as the air. Thus, we do not use this SURFEX option, but it could be tested in future, decoupled studies.*
*Concerning meltwater, when the overlying snow melts, meltwater is at 0C in the model (it carries no heat). Note that if meltwater does infiltrate into a layer which is below 0C, then freezing and latent heat release is modeled following Decharme et al. (2016).*

P10 L11 '$\tau_\alpha$. . . defining the runoff timescale from. . .' > $\tau_\alpha$ is important in this paper.

Then, I recommend that detail explanation of $\tau_\alpha$ are necessary here. For example, *** become larger with increasing $\tau_\alpha$ and so on.
*We have added a description to the text: "$\tau_\alpha$ controls the distribution of moisture, with larger values leading to a concentration of water at the debris-ice boundary and smaller values leading to a more even distribution. All values considered give an increase in water with depth, which is to be expected with the combination of gravity and the fact that debris clasts get finer with depth."*

P11 Figure 4 I recommend to change the unit of $\tau_\alpha$ in hour rather than in second, because in the text you have discussed the $\tau_\alpha$ in hour.
*FIXED.*

P12 Line 5 I recommend that all meteorological variables are described in the supplement.
*We considered this but decided to keep all information about the meteorological data in the main text, given its importance and the comments from reviewer 1.*

P12 L13 There are no description about how did you measure the debris temperature and debris density and porosity.
*These are added to the enhanced Section 2 "Field Site and Data."*

P12 L21 I estimate you used eq.(1) in Wagnon et al.(2009) here. But please draw the equation between -1 to +3 C. The equation has no simple relation. I can't imagine why you have to use the complicate relation. Or you have applied for the temperature range between 0 and +2 C ?
*We did, indeed, use the fifth order polynomial between -1 and 3 $^o$C, as given in Wagnon et al (2009), for the sake of completeness. We agree that it would have been*

*more logical to apply the equation between 0 and 2 °C only. The relationship between solid precipitation and temperature is complicated in the tails, as the reviewer's plot elucidated. Using a value of one from -1°C - 0°C and zero between 2°C and 3°C would be more straightforward but is highly unlikely to have any impact on simulated glacier melt given how close P is to 1 and 0 in these ranges.*

P12 L22 'minor adjustments based on SR50 measurements.' > There is no detail information on the adjustments. I can estimate that measurement of precipitation using Geonor T200B has relatively small error because it can measure weight of precipitation directly. On the other hand SR50 measure only surface elevation change, which is not equal to amount of precipitation. So, why you should adjust using SR50 ? *The SR50 ultrasonic depth gauge indicates whether or not snowfall has taken place at the field site. We use precipitation data from Pyramid and partition phase based upon AWS temperature. However, the SR50 may indicate snowfall when there was none recorded at Pyramid, and it may conversely indicate no solid precipitation when subfreezing temperatures at the West Changri Nup AWS coincided with recorded precipitation at Pyramid. Therefore, it is necessary to perform site-specific adjustments, which we now describe.*

P13 Table 2 You should write name of instrument at the 'Instrument' for example Pyranometer for shortwave radiation.
*FIXED.*

Campbell SR50 can not measure ablation or accumulation directly. It can measure only elevation change, then, you should assume snow or ice density to estimate accumulation or ablation.
*We added this distinction in the table caption.*

P14 Figure 5c It seems that many point of rainfall have been overlaid by those of snowfall. I think these two data should be shown separately. The label of x-axis should rotate 90 degree. Vertical writing for label of x-axis is easy to detect the exact location.
*We separated the plots of rainfall and snowfall and extended the extents of the y-axes to ensure that the x-axis ticks are visible.*

There are rainfall, but no snowfall during winter (from Jan 2014 to around March 2014). Is that possible? Please check.
*This figure has been FIXED.*

P15 L17 '2013 ablation' > There is no information of the observation of 2013 ablation. Location? Duration? Frequency of measurement ? How much is the debris thickness? How did you measure? And the location of ablation measurement were same with that of measurement of debris temperature ?
*This information has been clarified in and/or added to the enhanced Section 2 "Field Site and Data."*

P15 L18 'using an RMSE calculation to capture the magnitude of temperature.' >> I recommend to write them specifically.
*The minimum RMSE, for $\tau_\alpha =30$, was 1.99995 which is a shallow minimum given an RMSE range of 1.99995 – 2.08845. The RMSEs for $\tau_{max}$ varied even less, with a range of 0.05; accordingly, we did not find a compelling justification to have the value of this parameter differ from the timescale of sandy soil to drain to its field capacity. While the suggestion to include these values in the text is certainly an important one, we decided against highlighting them for fear that they would confuse the majority of readers and detract from the message.*
P16 Figure 6 I recommend that daily average data both observation and calculation are plotted in one graph. I cannot compare if those data are shown in different graph. Further, many points are overlapped in Fig. 6, then, daily average are necessary. And I also recommend that scatter plot of the observed and simulated debris temperature are valid to compare and analyze. And how much is the debris thickness? 12.5 cm ?
*It is a good idea to include a direct comparison of measured vs. modeled temperatures, and such figures have been added to the Supplement. We have updated the caption of this figure (formerly 6) to emphasize the reason for the figure (i.e. to show why we chose the tuning period we did) and to reiterate the debris thickness.*

P16 Figure 6 '(having insufficient snowfall to produce the observed persistent snow-cover)' > It might be possible that model overestimate the melt rate of snow cover. I think reconstruct of duration of snow cover during the melting season is significant (see main comment).
*Yes, there could potentially be errors associated with inaccurate reconstruction of snow cover; however, doing snow cover reconstruction and/or an analysis of it is out of the scope of the paper. See our response above concerning the snow cover discussion added to "Future Directions." Additionally, it is indeed true that if any of the simulated incoming components to the energy balance are too large, the model could potentially compensate (equifinality) by heating the surface too much and, thus, overestimating the melt rate of the snow cover. We added a statement that a snow cover simulation error could be responsible for the overestimation of sensible heat flux.*

P17 Figure 7 In the explanation, '(a) Temperatures in debris (top 12.5 cm)' I can't understand the 'top 12.5 cm'. This points location of interface between debris and ice? '... below the black line indicates...' > 'below'?
*This caption has been rewritten to avoid confusion.*

P17 L3 Figure A2 > Figure 2
*FIXED (Figure S2).*

P 18 Figure 8 I recommend to describe those data is simulated data.
*Clarification added to the figure caption.*

P20 Figure 10 There is no calculated data shown in Figure 8 (calculation of moisture integrated model). Why? It seems that the calculated ablation using moisture integrated model (Figure 8) have similar value with 'dry' and 'partially saturated'. And in this section (P20 L1-8), there is no discussion about the difference between three case calculated data and measured data. Measurement value was corresponded with the value of assuming fully saturated, not dry or partially saturated.
*Figure 8 is all model output, what we think this reviewer terms "calculated data," and the caption has been updated to reflect that. Figure 8 shows the "partially saturated" case, a point made clear in the caption to the table with which we replaced Figure 10. The "Wet versus Dry Debris" subsection contains an enhanced discussion of the distinction between the three cases and the comparison with measurements.*

P21 Figure 11 This is a minute thing, but as for color bar, 'No moisture' are colored by light red. I think the color should be white.
*FIXED.*

P22 Figure 12 I recommend to add cumulative melt (like Fig. 8).
*Our original manuscript stated that "cumulative flux used for melt" was part of figure 12. In the submitted version, we intended to show only latent heat fluxes and have removed the language that remained in the text from a previous revision.*

In the last sentence of the explanation, there are 'A greater latent heat flux corresponds to a lower conductive heat flux through...' > There are multiple latent heat fluxes in this system. Please write specifically.
*Rewritten as: "Greater latent heat fluxes are balanced with a lower conductive heat flux through the debris into the underlying glacier."*

P24 Table 3 '% Change' > What is the change ? Change of ablation ?
*Actually, it is melt. Table heading changed to "% Change in Melt."*

P28 L12 '75.3 ±20 cm (2012 − 2013) and 47.1 ±20 cm (2013 − 2014)' > Here, I have noticed that you have observed data not only 2012 − 2013 but also 2013 − 2014. Why you did not compare observed data and simulated data 2013 − 2014? Because you have tuned parameter 2013 − 2014? But, you have tuned parameter $\tau_\alpha$ fitting simulated debris temperature to observed debris temperatures. Not ablation amount. Observed data of ice melt under debris layer are very few, even though those data have large uncertainty.
*We converted this to a table with model output from both years and available point mass balance measurements. We erroneously compared model-computed melt to point mass balance in the original submission but have corrected this, along with reporting modeled mass balance components and both stake and SR50 depth gauge measurements.*

P29 L23 'more detailed assimilation of the snow rate with the SR50 data.' > Still I don't know the location of the measurement of ablation and debris temperature. Then, my comments might not appropriate. I would like to ask the surface temperature at AWS (SR50 measurement) can be assumed to be same condition with that at ablation measurement site? If you can assume same condition, you can analysis the albedo of AWS using observed SRs downward and upward and weather there is snow cover or

not.

*From the available data, we can, indeed, calculate whether there was presence or absence of snow. This is a great point and a methodology that could be employed in future work.*

P29 L24 'Snow is a strong insulator, and any error in simulated occurrence of snowfall will cause error in the surface temperatures and underlying debris temperature profile simulated by ISBA-DEB (e.g. Figure 6).' > I recommend to discuss about the snow cover during summer season in the discussion section. Because you have observed data of temperature of debris as shown in Fig.6.
*Please see our response above on the scope of the paper and what we added to the Discussion on the subject.*

P29 L26 'ISBA-DEB may be used to explore past or future changes in sub-debris melt. Reanalysis data, such as that of ERA Interim, provides all variables necessary to drive the model.' > ERA Interim do not include debris-thickness or particle size of debris, which is important for permeability (thermal conductivity) of debris and also surface roughness, those parameter have high sensitivity in ISBA-DEB model in your result.
*The model is driven by meteorological values only, and we have added a list of those exact variables to subsection 2.3 to prevent confusion.*

How will you apply the ISBA-DEB model to other debris-covered area? I recommend that you have to better to consider application to other debris covered part.
*The model can be applied to debris-covered areas for which debris physical properties (second section of Table A1) and continuous meteorological data (subsection 2.3) are available. We have bolstered the discussion about the applicability and transferability of ISBA-DEB to other debris-covered glaciers and included considerations on scaling the model from the point to glacier scale.*

---

## Author Comment (AC2) · 25 Jan 2020

**General comments:**
It is certainly a good advance to include the moisture explicitly in a model of sub-debris melt and modifying ISBA to do this seems like a good idea and something I would like to see published. The paper is well written on the whole and explains the model advance quite well. Its concisely written and the overall approach seems robust. There are likely still some parts of the model which are less like the reality in the debris (e.g. the overestimation of the sensible heat flux and possibly moisture being held in the lowest debris layer) but nevertheless this is an important step forward.
*First, we want to thank the reviewer for his/her thorough review, which greatly improved this manuscript. We have bolstered the discussion on the sensible heat flux overes-*

[Figure]

*timation, including by describing equifinality, and explained the moisture distribution and debris in greater detail. The concentration of moisture at the debris – ice transition matches both our own and others' (Collier et al., 2014; Nicholson and 10 Benn, 2012; Rounce and McKinney, 2014) observations but is, as we show, also a function of glacier slope (with flatter debris layers holding more moisture).*

In places the paper can lose clarity a little, and could do with explaining some parts of the model, and especially the collected field data more explicitly (see specific comments for details).

*We have added "Field Site and Data" before the model description (Section 3).*

Moreover, I think a key thing which is missing is the ability of the authors to really bring out the understanding of the interplay between the energy and moisture fluxes and the changes to the debris properties and how this affects melt overall.

*Our revised text highlights these relationships, particularly a central takeaway, "not only does a wet debris layer transfer heat less efficiently from its surface to its base than dry debris because of a decreased thermal diffusivity, but also it has less energy to transfer in the first place because of the other energy fluxes (mainly the surface latent heat) associated with the scenario."*

This could be strengthened in the results and discussion by better explaining the links between the debris properties and moisture content when explaining the results (improving some of the figures would help in this regard too). For instance, a clear explanation is missing of how the moisture content, thermal conductivity and heat capacity evolve on a daily and seasonal scale, of why moisture tends to be concentrated in the lowest layer and what the explicit differences are between the dry and saturated conditions.

*We strengthened our description of the links between the debris properties and moisture. For example, "During the summer, the glacier is melting, and the bottommost layer of debris is almost always saturated with liquid water, such that it has a thermal conductivity K of $\sim$1.16 W m$^{-1}$K$^{-1}$ (Table 1). The thermal conductivity of layer*

*13 changes little throughout the day; layer 1 shows the most variation in conductivity because its water content experiences the most variation (condensation followed by evaporation). A higher value means more water content while a lower value indicates dryness. The diurnal patterns in conductivity (Figure S4a) are replicated in specific heat capacity (Figure S4b) because both are functions of water content, and both thermal conductivity and heat capacity have higher values for the water-saturated debris in layer 13 than the drier debris in the overlying layers." (Section 5.1)*
*We have addressed the concerns listed, and details on how we revised the manuscript and figures can be found in the "Specific Comments" Section below.*

Also, although the future directions section compares the results with the broader literature, this could be strengthened by having a clearer uncertainty analysis (and hence clearer recommendations for where data and understanding is lacking) and a clearer idea about what steps would be needed to bring this modelling from the point to the glacier scale.
*Making a comprehensive uncertainty analysis, including Monte Carlo simulations, is beyond the scope of the paper, though such detailed studies have been undertaken for ISBA (Emery et al, 2016) and a SURFEX soil model (Garrigues et al, 2018). We believe that our sensitivity tests are sufficient for capturing uncertainties inherent to the model and uncertainties associated with the debris data paucity on West Changri Nup glacier. Accordingly, we changed the language surrounding the purpose of the sensitivity tests to convey their role and importance: "We performed sensitivity tests on the six parameters. . . to explore and quantify uncertainty associated with parameterizations in ISBA-DEB."*
*>>Emery, C., S. Biancamaria, A. Boone, P.-A. Garambois, B. Decharme, S. Ricci and M. Rochoux, 2016: Temporal variance-based sensitivity analysis of the large scale hydrological model ISBA-TRIP: Application on the Amazon basin. J. Hydrometeor., 17, 3007-3027.*
*>>Garrigues, S., A. Boone, B. Decharme, A. Olioso, C. Albergel, J-C. Calvet, S. Moulin, S. Buis, E. Martin, 2018: Impacts of the soil water transfer parametrization on*

*the simulation of evapotranspiration over a 14- year Mediterranean crop succession. J. Hydrometeor., 19 (1), 3-25.*

*Additionally, it is worth noting that modeling on the glacier scale was one of our initial goals, but our focus was on 1 dimension after deciding to adapt ISBA-DEB. At this stage, such scaling (point to glacier-wide) is not possible. Nevertheless, we believe that our study can help to inform development of a spatially distributed model through our findings about the most sensitive parameters controlling the melt at point scale. Such an assertion/sentiment has been added to the Conclusion.*

Furthermore, do you think that the findings determined for this glacier would hold for others, for instance with finer or thicker debris?
*The idea is certainly to apply ISBA-DEB to other debris covers, and we try to make the user flexibility (and model applicability) clear through Figure 3 and Table A1. We did not run ISBA-DEB for a debris cover of a different porosity or thickness, so we cannot include such details.*

At the moment the paper presents the new modelling (which is great to see), but I think it would really increase the usefulness of the paper if it could go beyond this a bit more to both better explain the findings and think more broadly about the consequences of those findings. I would also say that the key 'take home messages' of the paper are not so clear. I feel less sure about what the new insights are which the modelling has made possible, the authors could therefore make this much clearer in the results/discussion and conclusions.

*The main purpose of this paper is to present the new modeling; we are happy that the reviewer considers this interesting. In the revised version, we distill take home messages and highlight the paper's contribution to debris-covered glacier surface energy balance modeling. These are now summarized in our revised Conclusion, part of which reads, "In its simulations of West Changri Nup glacier, enhanced melt occurs below dry debris due to a combination of greater thermal diffusivity and little loss of energy to evaporation or sublimation. ISBA-DEB explicitly accounts for the*
*atmosphere-debris latent heat exchanges in the top (surface) layer of the debris only. The large difference in glacier melt below dry and saturated debris shows that latent heat is enormously important in removing energy from the system. Accounting for moisture in the conductive heat flux alone is insufficient when modeling melt under a debris-covered glacier. It is, therefore, an essential next step to examine and incorporate the latent heat exchanges of moisture at all depths in the debris.*

*ISBA-DEB provides a basis for developing a model that can be applied at the glacier scale by identifying not only the importance of atmospheric exchanges throughout the debris column but also the most sensitive parameters controlling the melt at point scale. In addition to using accurate roughness lengths, it is crucial to represent moisture sources and sinks correctly."*

One other point is the use of the Reid and Brock (2010) parameters for dry debris, when the values were measured under conditions when there was likely melting at the ice-debris interface, and so the values would be for partially saturated conditions.

*Great point. We assume that Reid and Brock (2010)'s value for Miage glacier's debris cover is applicable to Changri Nup glacier in the absence of (a) other observations and of (b) a straightforward way to adopt geological values for solid bedrock (reported for very high temperatures and pressures) to porous covers of sand, gravel, cobble, and boulder sized debris clasts. We amended the language in the caption to Table 1 to make it clear that we make an assumption about the applicability of Reid and Brock (2010)'s value as opposed to mis-cite the source. We discuss this assumption, cite other K values, and calculate thermal conductivity for gneissic debris in the Discussion. The text now includes, the following:*

*"A study on Miage glacier, Italy provided 0.94 W $m^{-1}K^{-1}$ as a starting point for thermal conductivity tests (Reid and Brock, 2010), though we varied the conductivity values throughout the range reported in the literature, 0.60 to 1.29 W $m^{-1}K^{-1}$ (Rounce et al, 2015). This was a particularly important sensitivity test to perform because, as noted in the caption to Table 1, the thermal values we assumed valid for dry debris on West*

*Changri Nup glacier were "effective" values reported for Miage glacier. As Brock et al. (2010) measured thermal conductivity in the ablation season, the reported K of 0.94 W m$^{-1}$K$^{-1}$ (Reid and Brock, 2010) was likely higher than that of perfectly dry Miage glacier debris since any moisture in the pore spaces would have had K = 0.57 W m$^{-1}$K$^{-1}$ (water) rather than K = 0.024 W m$^{-1}$K$^{-1}$ (air). Additionally, debris on Miage glacier (Italy) may have a dramatically different lithology than the debris on Changri Nup glacier (Nepal). Reports of debris conductivity on Khumbu glacier, adjacent to Changri Nup, include 0.85 W m$^{-1}$K$^{-1}$, 1.28 W m$^{-1}$K$^{-1}$ (Conway and Rasmussen, 2000), and 0.96 W m$^{-1}$K$^{-1}$ (Rounce and McKinney, 2014) and indicate that 0.94 W m$^{-1}$K$^{-1}$ is not inappropriate to apply to West Changri Nup. West Changri Nup's debris is most likely comprised of the sillimanite gneiss that forms its surrounding mountains (Searle et al., 2003). The USGS's "Thermal Properties of Rocks" (Robertson, 1988) gives a thermal conductivity of 2 W m$^{-1}$K$^{-1}$ (see Figures 3 and 16 therein). For debris with 39% porosity and air-filled pore spaces, a weighted-average K for dry debris is 1.2 W m$^{-1}$K$^{-1}$, which is within our tested range. Thermal conductivity is difficult to measure in the field, and it is not known how transferable the limited available measurements are to other debris covers and conditions. It is also not known whether a weighted average of bedrock and air thermal properties is a valid representation of porous debris. Accordingly, we intended to encompass the true value(s) in the range for which we tested ISBA-DEB's response."*

**Specific comments:**

Pg 2 Line 19: Reid and Brock don't necessarily assume 'dry' debris, just assume the debris characteristics are constant and the same as the average measured conditions. *Please see our response above including some of the text added to the manuscript.*

Pg 2 It would be worth mentioning Evatt et al. (2015) Glacial melt under a porous debris layer, somewhere in the introduction (likely in the paragraph beginning on line 18), given their inclusion of the evaporative heat flux at the bottom of the debris layer.
*This reference has now been incorporated into the text.*

Table 1: It would be better if the equations and the explanation in the caption could be taken out of this table and moved to Supplementary material.
*FIXED. Table 1 in the text includes no equations or references; all of that information is now part of a complementary table in the supplement.*

It is not clear where the values for ice and water conductivity and volumetric heat capacity are from (even if they are standard values).
*References for these values have been added.*

Given the importance of these parameters it would help to clearly explain the relation between them in the introduction.
*Added to the Introduction: "The efficiency with which a debris layer conducts heat is determined by its thermal diffusivity $\kappa$ ($m^2\ s^{-1}$). $\kappa$ is given by thermal conductivity K ($W\ m^{-1}K^{-1}$) normalized by the volumetric heat capacity ($J\ m^{-3}K^{-1}$), itself product of density $\rho$ ($kg\ m^{-3}$) and heat capacity $c$ ($J\ kg^{-1}K^{-1}$)."*

Also, it is unclear why the Reid and Brock (2010) value of thermal conductivity is used for dry debris, given that in the caption it is termed an 'effective' value from the time of measurements (this is confirmed in Brock et al, 2010). It is likely that this value is for a partially water saturated debris layer, rather than a dry one. Even if you decide this is a value for 'dry debris' then I think you would need to know the porosity of the debris on Miage Glacier to determine the 'debris' thermal conductivity so that you

could then calculate water and ice saturated debris. I am not convinced therefore that the calculation of the water and ice saturated debris is correct.
*See response to the similar comments above.*

Fully name the parameters, e.g. Thermal conductivity, Specific heat capacity.
*FIXED.*

Field site: it would be good to get an idea of the debris thickness, grain size and geology for West Changri Nup.
*We add a citation for Lejeune et al (2013), which contains a picture and description. We also include, "The debris is a granitic metamorphic mix, likely consisting of gneissic clasts eroded from the surrounding cliffs."*

Figure 1: This map could be improved. The text on the inset map is unreadable, and the inset needs a scale bar. The main pane is too zoomed in, it would be better to show the full outline of both glaciers and label them.
*All suggested changes made.*

Pg 5 Line 23: 'air temperature, humidity and surface fluxes. . .' Also, for clarity would be better to state the surface fluxes measured (incoming and outgoing short and longwave radiation).
*FIXED.*

Figure 2: Is the heat flux due to precipitation calculated?
*It is not. We have clarified this in the text.*
Pg 6 line 1: Is the surface debris temperature calculated, or taken from measure-ments?
*Neither. We have clarified this in the text: "The prognostic state variables are assumed to be located at the midpoint of each layer; accordingly, the uppermost simulated temperature is at 0.5 cm depth in the debris, not the surface."*

Pg 6 line 8 It is not clear what data is used for the spin up, 40 x the met data?
*Correct. FIXED.*

Pg 6 How many layers are for snow?
*We enlarged this detail in the model schematic and also added it to the text.*

Figure 3: Give the symbols for the input variables within this figure (especially the top right bubble), so it is absolutely clear what the input variables are. In the third box down, surely all the fluxes (not just latent heat) are calculated?
*FIXED. Yes all the fluxes are calculated; we aim to highlight differences between ISBA and ISBA-DEB.*

Pg 8 line 6: Refer here to the supplementary material where you give the full energy balance.
*FIXED.*

Equation 3: Should the wi not be wmin in the brackets?
*We do not understand this comment and have confirmed that equations (3) and (4) resemble those in the work cited.*

Pg 8 Line 27: 'where Lm is the. . .'
*FIXED.*

Also it would be useful to explain in words what each part of these equations represent. I'm a little unclear about the source and sink terms and how they represent the system.
*Clarification added to the text.*

Pg 9 Line 10: The lack of ice growth in subfreezing temperatures may suggest a lack of vapour transport, but what about in above freezing temperatures?
*Clarification added to the text.*

Pg 9 Line 14: How was the isothermal vapour conductivity derived, or what value was used?
*Reference added.*

Pg 9 Line 15: The observations of moisture transport, are these yours, or another author? It would be interesting to know of measurements of moisture transport if these are available. If based on your observations then that's fine but make this clear.
*We do not discuss observations of vapor transport but rather of water pooling at the ice-debris interface. We have clarified this in the text.*

Pg 9 Line 24 Should there not be brackets around wi/w (following Boone et al., 2000)?
*Good catch. FIXED.*

Pg 9 Line 32: It would be worth mentioning that debris tends to coarsen upwards (e.g. as mentioned by Reid and Brock, 2010).
*Added to the text.*

Pg 10 Line 6-7: Do you mean with depth of water or with depth in the debris? And why would it increase with depth (of debris, especially?)
*The runoff timescale increases with depth in the debris, as water content tends to be higher with greater depth in the debris. An explanation for this distribution has been added to the manuscript in two different places.*

Equation 8: I can't see $\Delta z_j$ or $w_j$ defined anywhere? (I think the subscript j just needs defined somewhere, unless I've missed it)
*FIXED.*

Figure 4: Is this depth of the debris or water depth on the y-axis?
*Y-axis label changed to "Depth from Debris Surface."*

Section 4: It would probably be more useful to have the forcing before the model, although I understand that you are presenting a new model method. It is just easier to follow.
The meteorological variables could easily be listed or put in a table (maybe refer to Table 2), there can't be that many. The section on in-situ meteorological measurements is lacking detail. How was the surface debris temperature situated (debris surface temperatures are easily overestimated), also give the depths of the sensors? How was ablation measured (with a UDG?) Give a reference for the ISBA-DEB being insensitive to the CO2 flux. The point about the precipitation should be moved to section 4.2 (in fact the section headings 4.1, 4.2 and 4.3 could easily be removed). How was debris density and porosity measured?
*The paper structure has been rearranged to include "In Situ Measurements" and*

*"Model Inputs" subsections within Section 2: "Field Site and Data," before the model
description (Section 3), including all information from former sections 4.1-4.3. Forcing
variables are now listed and details are added to the in situ measurements section
(including on debris temperature sensor depths and ablation measurements). We
changed the sentence about $CO_2$ to mention all SURFEX-required input.*
*SURFEX requires $CO_2$ flux because it includes a vegetation option we do not use.
Despite the options to include soil organics (Decharme et al, 2016) or soil carbon
(Delire et al., under revision), neither option is currently used for ISBA-DEB, the focus
of which is rocky debris and moisture.*
*>>Delire, C., Seferian, R., Decharme, B., Alkama, R., Calvet, J., Carrer, D., Gibelin,
A.-L., Joetzjer, E., Morel, X., and Rocher, M.: The global land carbon cycle simulated
with ISBA: improvements over the last decade, Journal of Advances in Modeling Earth
Systems, under revision.*

Pg 12 Line19: When does the precipitation dataset end? How was the precipitation at
Pyramid corrected (or give a reference that explains)?
*We added a reference (Sherpa et al, 2017) and clarified that the precipitation dataset
extends beyond the end of our study.*

Pg 15 Line 21 Give the range in error of $\tau_\alpha$ and $\tau_{\max}$ to show that the actual error
values were similar.
*The minimum RMSE, for $\tau_\alpha =30$, was 1.99995 which is a shallow minimum given an
RMSE range of 1.99995 – 2.08845. The RMSEs for $\tau_{max}$ varied even less, with a
range of 0.05; accordingly, we did not find a compelling justification to have the value
of this parameter differ from the timescale of sandy soil to drain to its field capacity.
While the suggestion to include these values in the text is certainly an important one,
we decided against highlighting them for fear that they would confuse the majority of
readers and detract from the message.*

Figure 6 it would be more useful to show the mean diurnal values, or at least only the time when both are compared within the same depth, with measured and modelled on the same panel. This would make it much easier to compare the measured and modelled outputs.

*It is a good idea to include a direct comparison of measured vs. modeled temperatures, and such figures have been added to the Supplement. We have updated the caption of this figure (formerly 6) to emphasize the reason for the figure (i.e. to show why we chose the tuning period we did).*

Figure 7 Text on these figures could be a little larger. On b) consider including the air temperature and debris surface temperature.

*Text size increased. We have considered adding air temperature and debris surface temperature to (b), but the figure became too noisy. Further, we wish to display model output, and the two suggested variables are both very different quantities (measurement and calculation, respectively, and not model output).*

Pg 17 Line 8 The above freezing temperatures propagating into the ice is shown better in Figure S2.

*We agree and refer instead to Figure S2 in the text for temperature propagation into the ice.*

Figure 8 Really Figure S2 is more useful than this figure, consider swapping the two.

*We considered this and think that Figure 8 is a more compelling figure for a non-modeler glaciologist and, as a result, have elected not to switch the figures. However, we have added a specific mention to S2 in the caption for 8. We leave it to the discretion of the editor whether to switch the figures (or to include both S2 and 8 in the*

*text).*

Page 18 The description of the change in the debris moisture content is a bit too terse. It would be helpful to describe this more thoroughly, especially as this is a key new function of the model. How does the thermal conductivity and heat capacity vary with the moisture content on a daily and seasonal scale? Also, explain why only the bottom layer is holding water.
*We added more descriptive language to Section 5.1.*

Figure 9 It would help to add side panels to both a) and b) of the diurnal average values for each layer (perhaps just during melting). This would help interpretation.
*We generated the suggested figures and added them to the Supplement as including them as part of Figure 9 made the figure too busy and difficult to interpret.*

Figure 10 This would be fine as a table, with the 2014 results given too. I am surprised that the results for the full model run with the integrated moisture content is not included too?
*We converted this to a table, submitted as a supplement along with this response, with model output from both years and available point mass balance measurements. We erroneously compared model-computed melt to point mass balance in the original submission but have corrected this, along with reporting modeled mass balance components and both stake and SR50 depth gauge measurements. The "full model run with the integrated moisture content" is, indeed, included; it is labeled, "partially saturated."*

Pg 20 Line 5 Compare the dry and saturated debris directly for clarity, i.e. the thermal diffusivity of dry debris is around double that of saturated debris, hence the higher melt

under dry debris.
*We added the suggested language to the text: "The fully saturated debris conducts heat that leads to melt with an efficiency of $\kappa = 3.572 \times 10^{-7}$ m$^2$ s$^{-1}$, while the dry debris has a diffusivity of $5.867 \times 10^{-7}$ m$^2$ s$^{-1}$; hence, more melt occurs below the dry debris in ISBA-DEB."*

Pg 20 I don't think the model includes the wind dynamics within the debris (unlike Evatt et al., 2015), would doing so change these results?
*That is correct. We discuss the fact that ISBA-DEB neglects advection throughout the text, though we have added an explicit mention that advection includes the wind dynamics modeled by Evatt et al. (2015).*

Figure 11 Again it would be helpful to show the mean diurnal patterns in each layer as a side panel, especially within the ablation season.
*Added to the Supplement and referenced in the caption for this figure.*

Figure 12 Consider changing the upper panel title to 'Debris surface latent heat flux from atmospheric exchange'. Also label panels a) and b) as for the others.
*FIXED.*

Page 22 Line 5 Including debris moisture does not significantly decrease sub-debris melt – compared to what? (the dry scenario?)
*Yes. Language amended.*

Pg 22 Line 8 Can you explain why only the lowermost layer holds water, is this to do with the runoff parameterisation or the debris porosity?
*We have added an explanation to the text: "The concentration of wetness at the debris*

*base is due both to the fact that debris coarsens upward (Reid and Brock, 2010) and to the permeability of the overlying debris (precipitation quickly moves through the debris until it reaches the impermeable ice surface)."*

Pg 22 Line 19 Figure 12 is missing the cumulative flux used for melt.
*Our original manuscript stated that "cumulative flux used for melt" was part of figure 12. In the submitted version, we intended to show only latent heat fluxes and have removed the language in the text from a previous revision.*

Pg 23 Line 3 In Figure 10 though it seems to show similar melt between dry and partially debris?
*This sentence does not appear in the revised manuscript, making this comment moot.*

Pg 23 Lines 15-27 Consider just citing the papers in Table 3 to shorten this section.
*In this section, all parameters are described in the text, not just the roughness lengths in lines 15 – 27. We elected to keep the section the way it was originally written, but, if brevity is at a premium, we can shorten it. We leave that to the discretion of the editor.*

Table 3 Just give the symbols rather than the shortened variable names in the first column. It would also be helpful to know the % change of the parameters, otherwise its tricky to determine which parameters the model is most sensitive too. I understand the overall rationale of using the literature values to give sensible ranges but this would help clarify the sensitivity of the model.
*Symbols FIXED, and we now give % of physically plausible range in the value of the parameter to indicate impact on the results.*

Pg 24 Line 1 For the extreme values tested, is this with the other parameters at their *

value?
*Yes; caption updated.*

Pg 25 Lines 1-3 A more thorough description of which variables the model is most sensitive to is needed here (as mentioned would help to show the % change in each variable) and an idea of which ones are most likely to be well or not well known. Then suggest where work should concentrate to improve the knowledge of the most sensitive variables, including of their spatial distribution.
*We have enhanced the Sensitivity Tests section by clearly stating which parameters the model is most sensitive to, mentioning when a parameter is difficult to measure or otherwise constrain, and suggesting where to focus future work. We updated the table with the suggested %s.*

Pg 27 Lines 6-16 This very high surface temperature could be indicative of the debris properties not being correct, as the model might be compensating by increasing the debris surface temperature. I am presuming here that the debris surface temperature is modelled (I think it is, although I am not sure if this explicitly stated).
*Indeed, if any of the simulated incoming components to the energy balance are too large, the model could potentially compensate by overestimating the debris surface temp. In the text, we have added that there may be an error compensation (equifinality problem), where multiple parameter sets provide equally good model outputs.*

Pg 28 Lines 1-2 Here it would be useful if the authors could comment on what they think would happen over a real debris-covered surface where there is a combination of steep and shallow slopes. What would likely be the overall affect? On line 1 specifically the authors mention' overlying flatter glaciers' which is a strange term, say specifically flatter slopes, as the slope angle will vary across the glacier surface (especially for debris-covered glaciers).

*Phrasing FIXED, and we have added a discussion of applying ISBA-DEB to an entire debris cover.*

Pg 28 Line 11 what ratio are you referring to here?
*Ratio of momentum to thermal roughness lengths. This has been clarified in the text.*

Pg 28 Line 33 Explain why moisture deep in the debris is less prone to evaporation. (Possibly because of lower wind speeds and cooler temperatures at depth, but explain this based on your results).
*We have added the explanation: "in contrast to soil, supraglacial debris has a higher permeability and lower evaporation rate. A lower evaporation rate is consistent with the fact that debris stores moisture at depth. Moisture deep in debris is less prone to evaporation than moisture on the surface since permeability (and, thus, ease of air flow) generally decreases into the debris although some does evaporate."*

Pg 29 Line 23 'snow melt rate' ? Also consider 'measured snow melt' instead of 'the SR50 data' just so it is clear what you are meaning without readers knowing your sensor names.
*Clarifying language added.*

Table A1 I still have concerns over using the Reid and Brock (2010) values of debris thermal conductivity and heat capacity for dry debris.
*See above.*
There are no values given for the matric potential at saturation or b (what is this?). I understand they are predicted values, but does this mean they change over time if one value isn't given?
*Description of b added to subsection 3.3.3 with the introduction of PTFs. Values for b*

*and matric potential at saturation are calculated and added to Table A2.*

**Supplementary Material:**
I might have missed it but please refer to the energy budget section in the main paper.
*FIXED.*

Also, it would be helpful to include the equations for the sensible and latent heat fluxes, since these can vary between models.
*We have added a citation.*

Pg 3 Equation 13 LEgu I think should be LEgi (or vice versa) to match the text. C8
*FIXED.*

Pg 3 Equation 15 LEgf I think should also be LEgi? I can't see Wn defined, should this be the change in water stored in snow (Ws)?
*FIXED.*

Pg 3 Line 15 is Ps the melted snow in water equivalent?
*It is the snowfall rate in water equivalent, noted on the same page.*

Figure S2 This is a useful figure and I would suggest including it in the main text.
*See above for rationale behind keeping it in the Supplement.*

It would be useful to have the rain and snowmelt also in mm (to match glacier melt).

*FIXED.*

Rename Layer 13 as the ice surface (rather than glacier surface).
*FIXED.*

Write 'Temperature' rather than just T in the legend, and say ice content and water content, rather than just ice and water, for clarity.
*We have added these clarifications to the caption, as adding them to the legend made the figure too cluttered.*
Ideally the middle four panels should have the same y-axis scale for comparison. Alternatively show all 4 layers of water content together in a pane, with separate panes for ice content and temperature.
*We agree that consistency is important but found that the alternative suggestions do not display the model behavior as well. We have added a note to the figure caption pointing out the larger y-axis range of panel 5.*

**Technical corrections:**
Pg 1 line 12 'debris-covered'
*FIXED.*

Pg 4 Line 3 and line 9 'debris-covered'
*FIXED.*

Pg 4 line 9: 'mostly composed of clean ice.'
*FIXED.*

Pg 12 Line8: delete 'dot in'. 'AWS data was measured half hourly from the 6th December 212 15:00 – 28 November 2014 13:30 local Nepal time, and provided'
*This sentence was rewritten responding to other reviewer feedback making this comment moot.*

Pg 15 Line 18 'measured ones from'
*FIXED.*

Pg 17 Line 2 'Figure S2'
*FIXED.*

Pg 25 last paragraph, join this to the next paragraph.
*We understand that the second paragraph in question is only a sentence in length. We decide to keep it separate because it concerns a different subject.*

Pg 27 Line 24 'using an eddy correlation approach'.
*FIXED.*

Pg 28 Line 4 'thermal conductivity'
*FIXED.*

Pg 28 Line 5 'varied measurements of roughness length' (I am presuming here)
*FIXED.*

Pg 28 Line 12 'Measured ablation over the two years modeled' Strange phrasing here,

reword.
*Sentence omitted in revised version.*

Pg 28 Lines 12-19 Consider shortening this section.
*This has been shortened to a focus on temperature sensors.*

Pg 28 Line 30 'in comparing their two data collection sites,'
*FIXED.*

Pg 29 Line 20 'It also computes glacier melt'
*FIXED.*

Figure S1 Add a y-axis title 'Energy flux (W m-2)'
*FIXED.*

Please also note the supplement to this comment:
https://www.the-cryosphere-discuss.net/tc-2019-168/tc-2019-168-AC2-
supplement.pdf

**Supplement:**

| Mass Balance Components (mm w.e.) | Completely Dry Scenario (mm w.e.) | Partially Saturated Scenario (mm w.e.) | Fully Saturated Scenario (mm w.e.) | Measurements (mm w.e.) |
|---|---|---|---|---|
| 5/12/2012 to 02/12/2013 | | | | |
| Cumulated solid precipitation | 289 | 289 | 289 | |
| Melt | 1241 | 1237 | 771 | |
| Sublimation | 40 | 40 | 220 | |
| Evaporation | 0 | 81 | 549 | |
| Ablation | 1281 | 1359 | 1540 | |
| Point mass balance | -992 | -1069 | -1251 | -1080 (at SR50) -753 (at stake) |
| 02/12/2013 to 28/11/2014 | | | | |
| Cumulated solid precipitation | 368 | 368 | 368 | |
| Melt | 975 | 983 | 605 | |
| Sublimation | 15 | 15 | 441 | |
| Evaporation | 0 | 66 | 553 | |
| Ablation | 990 | 1064 | 1599 | |
| Point mass balance | -622 | -696 | -1231 | N/A |
| 09/04/2014 to 19/07/2014 | | | | |
| Cumulated solid precipitation | 142 | 142 | 142 | |
| Melt | 500 | 495 | 267 | |
| Sublimation | 8 | 8 | 338 | |
| Evaporation | 0 | 47 | 362 | |
| Ablation | 508 | 550 | 967 | |
| Point mass balance | -366 | -408 | -825 | -760 (at SR50) |

**Table 3.** Mass balance components of three model runs (dry debris, partially saturated debris, and fully saturated debris) compared with the available measured point mass balance. In order to achieve fully dry debris, all rain and melted snow was assumed to run off immediately (frozen snow could persist and sublimate; hence, there is zero evaporation but sublimation commensurate with that of the partially saturated scenario. The fully saturated scenario has debris water that sublimates when a snow cover is absent in sub-freezing temperatures). The measurements come from a bamboo stake, which carries an uncertainty of $\pm200$ mm w.e. (Vincent et al., 2016), and SR50-detected surface height changes using a snow density of 200 kg m$^{-3}$. In 2014 the stake broke and SR50 was operational from only 09/04/2014 to 19/07/2014. Dates are in dd/mm/yyyy format. *Runoff has not been taken into account for this comparison.

---

## Author Response (AR1)

[revised manuscript text omitted]

(a)

(b)

**Figure S4.** 2013 monsoon season (JJAS) mean diurnal values in local Nepal time (LT) for water (top panels) and ice (bottom panels) for the (a) partially and (b) fully saturated scenarios shown in Figures 10a and b, respectively. Only three layers are shown for figure clarity: debris top (0.5 cm), middle (6.5 cm), and base (12.5 cm).

[Figure]

**Figure S5.** Thermal conductivity (a) and volumetric heat capacity (b) of debris top (0.5 cm), middle (6.5 cm), and base (12.5 cm) layers throughout the time period used for this paper, with the 2013 monsoon season (JJAS) mean diurnal values for each (c, d)

**Sensitivity**

[Figure]

**Figure S6.** Cumulative glacier melt over the December 2012 – November 2014 period, run with different values of key parameters to test ISBA-DEB's sensitivity. An asterisk in legends indicates values used in the run demonstrating model behavior (Section 5.1), and sensitivities are quantified relative to melt simulated with these values in Table 4.

---

## Author Response (AR2)

Dear Tobias,

Thank you very much for your careful reading of our revised manuscript. Below we specify how we have addressed each of the minor revisions. We hope you find the present manuscript to your satisfaction; please let me know if you have further concerns. Sincerely,

Alexandra

Reviewer 1 asked whether you think that the findings would hold true for finer or thicker debris. I recommend to add at least a short discussion about the applicability instead of just refereing Fig. 3 and Table A1.

We have added the following to the Future Directions section:

"Simulating neither evaporation nor advective heat transfer within the debris accounts for the relative melt modeled under 12.5 cm of wet, dry, and partially wet debris (Section 5.2, Table 3). While ISBA-DEB can accommodate any debris thickness, so long as the entire debris-glacier column is discretized into 20 or fewer layers, we expect sub-debris melt values calculated by the model in its current version to match measurements of melt under thinner debris more closely than measurements under thicker debris. The effect of neglecting evaporation deeper than the surface is decreased for thinner debris, as is the effect of neglecting advection within the debris. With thicker debris – or, more generally, with debris covers in which sub-surface evaporation and/or advection are more favorable – we expect the departures from measured sub-debris melt values due to these unrealistic assumptions in ISBA-DEB to be more pronounced."

With respect to swapping Figures 8 and S2: This is a difficult decision and both figures are interesting. Ideally, I would like to see both in the manuscript. However, you have already quite a lot of figures and should not increase the number of figures. Maybe you can move another one to the supplement such as Fig. 5 which can easily described in words?

We moved Fig. 5 to the Supplement and have Figures 8 and former S2 (now 7) in the main text.

With respect to the comment on just citing the papers in table 3. I favour in general brevity and this manuscript is already relatively long. However, to be consistent in style and depth of information iI am in line to keep the section as it is.

I am not satisfied with the reply to the comment by reviewer 2 on P29 L23. Please make sure that the location of the measurements are clearly indicated. See here also my comment below on the caption for table 3.

We added text in Section 2.2 to clarify the proximity of these measurements to the AWS location:

- "ultrasonic depth from an SR50 adjacent to the AWS"
- "four resistance temperature probes installed at distributed depths (5, 7.5, 10, and 12.5 cm) in the 12.5 cm thick debris 40 cm from the AWS"
- "an 8-m long bamboo mass balance stake installed 5 m uphill (west) of the AWS"

Minor technical/stylistic comments:

P. 8L3: The link does not work properly, exclude "):". I suggest to add the full list also in the supplement as experience shows that links to websites can be outdated in a while.

We confirmed that the link works and also added the list in the Supplementary Materials.

P12L3f: The sentence is not clear. Please improve.

We split this long sentence into two sentences to improve clarity.

P21: Caption Table 3: The caption is very long and contains important information about data measurements, methods and uncertainty. This info should be presented in the respective section. I would omit the info about the data format as TC is a European journal and the order is very clear from the numbers like 19/07/2014

We moved the pertinent information in the caption to the text and removed the information about the date format.

P20 last sentence and P22 1st sentence. Here you refer in two subsequent sentences to the table which is not the best style. "Table 3 shows the three computed point mass balance values for 2013 and 2014, as well as available measurements." "The glacier under completely dry debris melts significantly more than the glacier situated under fully saturated debris in all three periods displayed in Table 3." I would add the years in the caption only and for the 2nd sentence simply refer to the table in

brackets instead of writing "displayed in".

We added the years to the caption, and the text surrounding the table now reads:

"In order to achieve fully dry debris, all rain and melted snow was assumed to run off immediately, although solid precipitation (snow) could persist and sublimate. Table 3 shows the three computed point mass balance values (not accounting for runoff) compared to available measurements. A bamboo stake, which carries an uncertainty of $\pm200$ mm w.e. (Vincent et al., 2016) and SR50-detected surface height changes, using a snow density of 200 kg m$^{-3}$, supply the measurements. In 2014 the stake broke, and SR50 was operational from only 09/04/2014 to 19/07/2014.

Important characteristics of Table 3 include the dry debris' zero evaporation but sublimation commensurate with that of the partially saturated debris; this results from the assumption that rain and melt run off instantaneously while SURFEX still models a snow cover that can sublimate. The sublimation computed in the fully saturated scenario is a sum of snow sublimation and debris water sublimation that occurs when a snow cover is absent in sub-freezing temperatures.

The glacier under completely dry debris melts significantly more than the glacier situated under fully saturated debris in all three periods (Table 3)."

P32L6: The reference Scherler et al. (2018) provides information about a percentage of debris cover on the global glacier and would an appropriate citation for the relevance of the study but not really for increasing debris cover. I would add here Kirkbride and Deline (2013) as also cited in Scherler et al. (2018) as evidence for increasing debris cover. You might also be interested to read in this regard Tildize et al. (2020).

Reference changed.

[revised manuscript text omitted]

**Table S1.** Thermally relevant properties of dry debris, in which interstitial pore spaces are filled with air; water-saturated debris; and ice-saturated debris of porosity ($\phi$) = 0.39. Air density is a function of elevation, air temperature, and air moisture. In the equation for air density, $\rho_{air} = P/(R_d \times T_v)$, P is pressure (Pa), $R_d$ is the gas constant for dry air ($\sim$ 287 J kg$^{-1}$ K$^{-1}$), and $T_v$ is the virtual temperature (K). Thermal conductivity presented by Reid and Brock (2010) is an "effective" value, from measurements, that is a function of debris' unspecified porosity and any moisture content at the time of measurement (Collier et al., 2016). Brock et al. (2010) used a published value of specific heat (948 J kg$^{-1}$ K$^{-1}$). We assume that these values of thermal conductivity and heat capacity (here listed for "dry debris") are valid for dry debris on West Changri Nup glacier and subsequently perform sensitivity tests. Note that diffusivity is conductivity normalized by volumetric heat capacity. Values for which no references are listed are the standard values used by SURFEX (LeMoigne, 2018).

**Model Inputs**

The list of continuous meteorological variables required to run SURFEX (found at *www.umr-cnrm.fr/surfex/spip.php?article215*) is reproduced below:

- Tair(time,yy,xx) : Atmospheric temperature (K)

- Qair(time,yy,xx) : Atmospheric humidity (kg kg$^{-1}$)

- PSurf(time,yy,xx) : Atmospheric pressure (Pa)

- Rainf(time,yy,xx) : Rain (kg m$^{-2}$s$^{-1}$)

- Snowf(time,yy,xx) : Snow (time,yy,xx) (kg m$^{-2}$s$^{-1}$)

- Wind(time,yy,xx) : Wind speed (m s$^{-1}$)

- Wind_DIR(time,yy,xx) : Wind direction (degrees from N, clockwise)

- LWdown(time,yy,xx) : Long-wave radiation (W m$^{-2}$)

- DIR_SWdown(time,yy,xx) : direct short-wave radiation (W m$^{-2}$)

- SCA_SWdown(time,yy,xx) : diffuse short-wave radiation (W m$^{-2}$)

- CO2air(time,yy,xx) : near surface CO2 concentration (kg m$^{-3}$)

[revised manuscript text omitted]

**Evolving Debris Properties**

805

[Figure]

**Figure S3.** Diurnal means of measured (blue) and modeled (red) debris temperatures for the calibration period marked by vertical lines in Figure 5.

[Figure]

**Figure S4.** 2013 monsoon season (JJAS) mean diurnal values in local Nepal time (LT) for water (top panels) and ice (bottom panels) for the (a) partially and (b) fully saturated scenarios shown in Figures 10a and b, respectively. Only three layers are shown for figure clarity: debris top (0.5 cm), middle (6.5 cm), and base (12.5 cm).

[Figure]

**Figure S5.** Thermal conductivity (a) and volumetric heat capacity (b) of debris top (0.5 cm), middle (6.5 cm), and base (12.5 cm) layers throughout the time period used for this paper, with the 2013 monsoon season (JJAS) mean diurnal values for each (c, d)

**Sensitivity**

[Figure]

**Figure S6.** Cumulative glacier melt over the December 2012 – November 2014 period, run with different values of key parameters to test ISBA-DEB's sensitivity. An asterisk in legends indicates values used in the run demonstrating model behavior (Section 5.1), and sensitivities are quantified relative to melt simulated with these values in Table 4.